# LARP1 binds ribosomes and TOP mRNAs in repressed complexes

James A Saba [ID][1,2,5], Zixuan Huang [ID][3,5], Kate L Schole[1,2], Xianwen Ye[3], Shrey D Bhatt[1,2], Yi Li[3], Winston Timp[1,4], Jingdong Cheng [ID][3✉] & Rachel Green [ID][1,2✉]

## Abstract

**Terminal oligopyrimidine motif-containing mRNAs (TOPs) encode all ribosomal proteins in mammals and are regulated to tune ribosome synthesis to cell state. Previous studies have implicated LARP1 in 40S- or 80S-ribosome complexes that are thought to repress and stabilize TOPs. However, a molecular understanding of how LARP1 and TOPs interact with these ribosome complexes is lacking. Here, we show that LARP1 directly binds non-translating ribosomal subunits. Cryo-EM structures reveal a previously uncharacterized domain of LARP1 bound to and occluding the mRNA channel of the 40S subunit. Increased availability of free ribosomal subunits downstream of various stresses promote 60S joining at the same interface to form LARP1-80S complexes. Simultaneously, LARP1 engages the TOP via its previously characterized La/PAM2 and DM15 domains. Contrary to expectations, ribosome binding within these complexes is not required for LARP1-mediated TOP repression or stabilization, two canonical LARP1 functions. Together, this work provides molecular insight into how LARP1 directly binds ribosomal subunits and challenges existing models describing the function of repressed LARP1-40S/80S-TOP complexes.**

**Keywords** TOP mRNA; LARP1; Ribosome; Translation; Cryo-EM
**Subject Categories** Signal Transduction; Translation & Protein Quality

## Introduction

Cells have evolved mechanisms to regulate ribosome synthesis in accordance with cell state. In mammals, a primary way this is accomplished is through the coordinate regulation of mRNAs encoding ribosomal proteins, all of which begin with a +1 cytidine nucleotide followed by 4–15 pyrimidines, termed the terminal oligopyrimidine motif (TOP motif) (Meyuhas and Kahan, 2015). Considerable evidence has converged on La-related protein 1 (LARP1)—a multi-domain conserved RNA binding protein (Bousquet-Antonelli and Deragon, 2009)—as a key regulator of TOP motif-containing mRNAs (TOPs). LARP1 binds the TOP motif and represses and stabilizes TOPs, most notably under conditions of mTOR inhibition (Fonseca et al, 2015; Lahr et al, 2015, 2017; Philippe et al, 2018; Gentilella et al, 2017; Philippe et al, 2020; Fuentes et al, 2021).

Structural and biochemical efforts have established key functional properties of three domains of LARP1. The N-terminal La and PAM2 domains bind polyA sequences and polyA-binding protein complex 1 (PABPC1), respectively (Al-Ashtal et al, 2021; Kozlov et al, 2022, 2024; Mattijssen et al, 2021), while the C-terminal DM15 domain has high specificity for the TOP motif in the context of an m[7]G cap (Lahr et al, 2015, 2017). Through these functional domains, LARP1 is thought to bind a large number of cellular mRNAs (Smith et al, 2020), presumably via its polyA- and PABPC1-binding activity, but to specifically engage TOPs via its DM15 domain. Beyond these structured functional domains, the majority of LARP1 protein is characterized as disordered, and it remains unclear whether other functional domains exist.

Surprisingly, recent evidence demonstrated that repressed LARP1-TOP complexes are associated directly with ribosomes. LARP1 and TOPs were shown to sediment with 40S ribosomes (dubbed the "TOP-40S") during normal growth conditions (Gentilella et al, 2017) or with 80S ribosomes (dubbed the "TOP-80S") when mTOR is inhibited (Hong et al, 2017; Fuentes et al, 2021; Schneider et al, 2022). However, the function, regulation, and mechanistic underpinnings of these interactions have been debated. One study suggested that the TOP-80S reflects single-ribosome (monosomal) translation that protects TOPs and permits their rapid reactivation (Schneider et al, 2022). Other studies have argued for a direct role in TOP stabilization conferred by 40S subunit binding to these complexes (Gentilella et al, 2017; Fuentes et al, 2021). Because the molecular and mechanistic underpinnings of these interactions have remained poorly defined, it has been difficult to definitively test these models.

Here, we define the molecular nature of LARP1 and TOP association with ribosomes in repressed complexes. Through biochemical and structural analysis, we demonstrate that LARP1 binds non-translating 40S ribosomal subunits through a previously

[1]Department of Molecular Biology and Genetics, Johns Hopkins University School of Medicine, Baltimore, MD 21205, USA. [2]Howard Hughes Medical Institute, Johns Hopkins University School of Medicine, Baltimore, MD 21205, USA. [3]Minhang Hospital & Institutes of Biomedical Sciences, Shanghai Key Laboratory of Medical Epigenetics, International Co-laboratory of Medical Epigenetics and Metabolism, Fudan University, Dong'an Road 131, 200032 Shanghai, China. [4]Department of Biomedical Engineering, Johns Hopkins University, Baltimore, MD 21218, USA. [5]These authors contributed equally: James A Saba, Zixuan Huang. ✉E-mail: cheng@fudan.edu.cn; ragreen@jhmi.edu

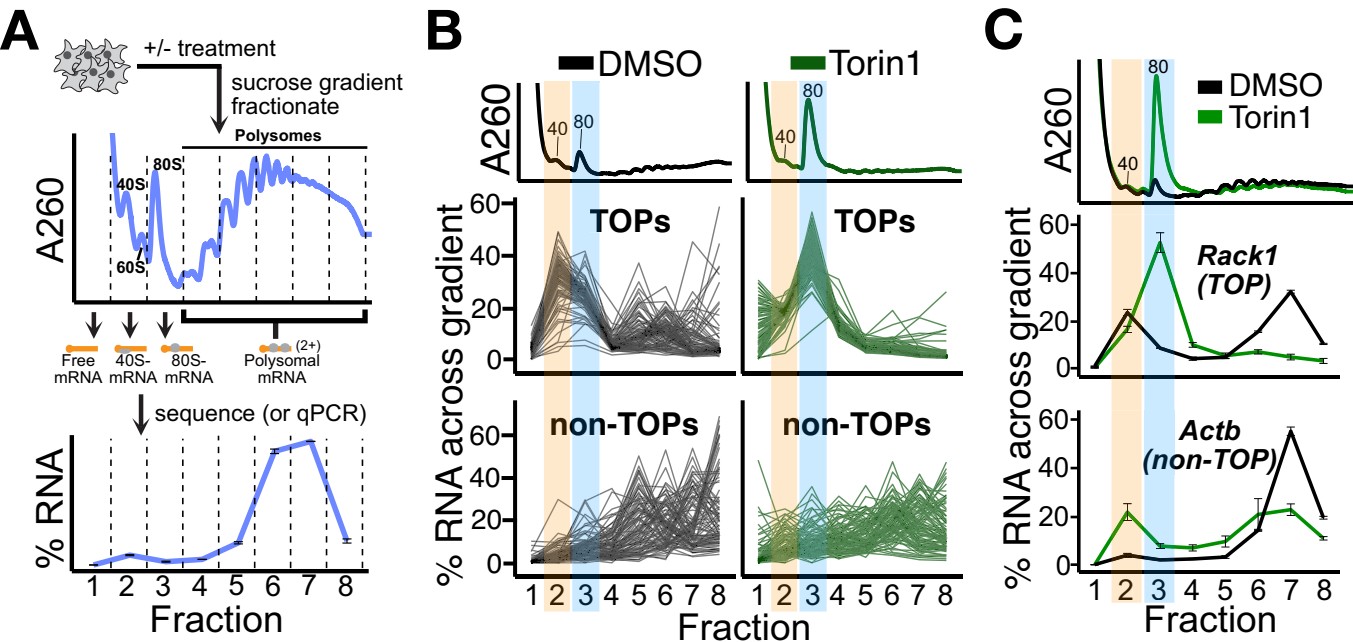

**Figure 1. System to study TOP association with 40S and 80S ribosomes.**

(A) Schematic of experimental setup. Cell lysates were separated by ultracentrifugation through a sucrose gradient and fractionated. The amount of each RNA in each fraction is presented as a percentage of the whole across the gradient, as shown for a representative mRNA. (B) U266B1 cells were treated with DMSO or 300 nM Torin1 for 1h, lysed, and processed as shown in (A), followed by PCR-cDNA nanopore sequencing. A260 traces for each treatment are shown in the upper panels. Each line represents the fractional distribution of a single mRNA across the gradient. Top: 96 annotated TOPs are plotted ±Torin1. Bottom: A random sample of 96 non-TOPs are plotted ±Torin1. Orange and blue highlights correspond to the TOP-40S and TOP-80S, respectively. (C) HEK293T cells were treated with DMSO or 300 nM Torin1 for 1h and processed as shown in (A), followed by qPCR against genes of interest. Orange and blue highlights correspond to the TOP-40S and TOP-80S, respectively. For (A–C), "40", "60", and "80" designations correspond to the canonical 40S, 60S, and 80S peaks by $A_{260}$. For qPCR plots, error bars are centered at the average and represent the SD from two to four technical replicates. Source data are available online for this figure.

uncharacterized domain which we call the ribosome binding region (RBR). In further characterization, we demonstrate that the TOP-80S comprises non-translating, weakly-associated 40S and 60S subunits bound directly to LARP1 protein, and not to the TOP itself. Surprisingly, ribosomal subunit binding within these complexes is not required for TOP repression or stabilization, challenging existing models. Our observations provide molecular insight into how LARP1 directly binds free ribosomal subunits and stimulates forward progress in understanding the regulation of TOPs.

## Results

### A system to study TOP association with 40S and 80S ribosomes

To study the association of TOPs with 40S and 80S ribosomes, we set up an assay in which we treated cells with either DMSO (control) or the mTOR inhibitor Torin1 (Thoreen et al, 2009), fractionated cell lysates across a sucrose gradient and followed the mRNA by performing nanopore sequencing of mRNAs isolated from each fraction (Fig. 1A). Normalization to a spike-in mRNA enabled us to quantify the percent distribution of each mRNA species in each fraction across the gradient and thereby assess

mRNA association with 40S subunits, 80S ribosomes, or with polysomes (multiple ribosomes engaged on an mRNA).

As expected, both mRNA length and CDS length were correlated with deeper sedimentation in the gradient, indicative of more ribosomes loaded on the mRNA (Appendix Figs. S1A–D). In DMSO, TOPs sediment in a bimodal distribution as previously observed (Meyuhas et al, 1987) (Fig. 1B; Appendix Fig. S1E): one population sediments in sub-polysomal fractions, representing translationally repressed mRNAs associated with 40S subunits (Gentilella et al, 2017; Fuentes et al, 2021), and the other population sediments in polysomal fractions, representing highly-translated mRNAs. In contrast, non-TOPs are more uniformly distributed across fractions and do not feature a bimodal distribution. Upon Torin1 treatment, TOPs redistribute en masse to the 80S fraction while the non-TOPs shift to somewhat lighter fractions but remain mostly within the polysomes (Fig. 1B). These data agree with previous reports demonstrating stronger translational repression of TOPs compared to non-TOPs with Torin1 (Thoreen et al, 2012; Philippe et al, 2020) as well as the redistribution of TOPs from 40S-to 80S-ribosome association upon Torin1 treatment (Hong et al, 2017; Fuentes et al, 2021; Schneider et al, 2022). Importantly, qPCR for individual TOPs recapitulated the same patterns of redistribution of TOPs and non-TOPs upon Torin1 treatment (Fig. 1C; Appendix Fig. S1F). These data define a robust system for interrogating the association of TOPs with 40S and 80S ribosomes.

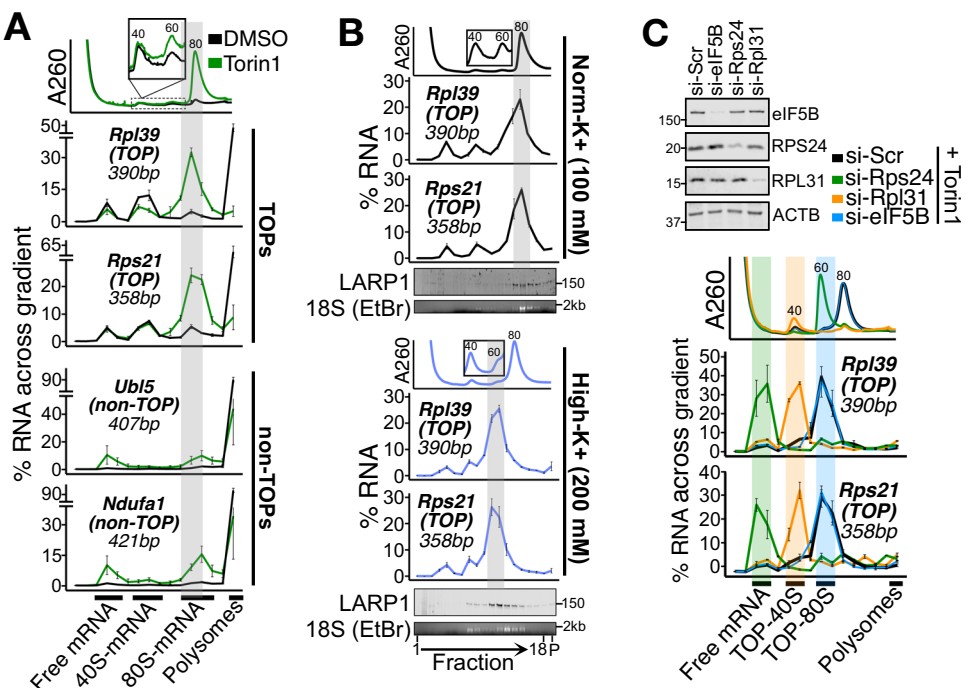

**Figure 2. Characterization of the TOP-80S complex.**

(A) HEK293T cells were treated with DMSO or 300 nM Torin1 for 1 h, and lysates fractionated on 15–35% sucrose gradients containing 100 mM KOAc followed by qPCR against genes of interest. Gray highlight corresponds to the TOP-80S. (B) HEK293T cells were treated with 300 nM Torin1 for 1 h, and lysates fractionated along 15–35% sucrose gradients containing either 100 mM KOAc (Norm-K$^+$) or 200 mM KOAc (High-K$^+$) followed by qPCR against genes of interest. Gray highlight corresponds to the TOP-80S. Western blots against LARP1 and EtBr gel for 18S rRNA from the same samples are presented below qPCR traces. (C) HEK293T cells were treated with siRNAs targeting scrambled (si-Scr), Rps24, Rpl31, or eIF5B followed by 300 nM Torin1 for 1 h (western blots demonstrating knockdown efficiency are shown). Lysates were fractionated along 15–35% sucrose gradients containing 200 mM KOAc (High-K+) followed by qPCR against genes of interest. Green, orange, and blue highlights correspond to free mRNA, TOP-40S, and TOP-80S, respectively. For (A–C), "40", "60", and "80" designations correspond to the canonical 40S, 60S, and 80S peaks by A$_{260}$. For qPCR plots, error bars are centered at the average and represent the SD from two to four technical replicates. For (A–C), biological replicate experiments are shown in Appendix Fig. S2A, C, D, respectively. Source data are available online for this figure.

## TOP-80S ribosomes are weakly-associated and non-translating

To better resolve the messenger ribonucleoprotein (mRNP) complexes in the sub-polysomal fractions, we ultracentrifuged cellular lysates over lower percentage sucrose gradients and fractionated more finely. Because mRNA length was correlated with deeper sedimentation in the gradient (Appendix Fig. S1A,B), we probed for endogenous mRNAs whose annotated size of <500 bp has a minimal influence on mRNP sedimentation. Hereafter we refer to these as "spread" gradients. This protocol allowed us to identify three distinct populations of TOPs in the sub-polysomal region of the gradient: (1) Free mRNA, (2) 40S-mRNA (TOP-40S), and (3) 80S-mRNA (TOP-80S) (Fig. 2A; Appendix Fig. S2A). We also extracted material that sedimented to the bottom of the ultracentrifuge tube to estimate the proportion of each mRNA in polysomes and, therefore, account for the full cytoplasmic distribution of a given mRNA species. As seen previously, TOPs (Rpl39 and Rps21) exit polysomes and sediment with 80S monosomes after Torin1 treatment while non-TOPs (Ubl5 and Ndufa1) are more resistant (Fig. 2A). In accordance with previous studies (Schneider et al, 2022; Fuentes et al, 2021), the

TOP-80S fails to form in LARP1-KO cells, confirming that LARP1 is a critical component of this complex (Appendix Fig. S2B).

Because of the high resolution of these gradients, we noted that the TOP-80S sedimented with the left edge of the A$_{260}$ 80S peak (Fig. 2A). We therefore wondered whether the ribosome complex forming with TOPs might be qualitatively different from other 80S monosomes that form under Torin1 treatment. We treated cells with Torin1 and compared the sedimentation of TOPs in gradients containing 100 mM ("norm-K$^+$") or 200 mM ("high-K$^+$") KOAc. While TOPs sediment with the left edge of the 80S peak in norm-K$^+$ gradients, they sediment even further left (closer to the 60S peak) in high-K$^+$ gradients (Fig. 2B; Appendix Fig. S2C). Based on past literature, this finding likely reflects ribosomal subunit splitting in high-K$^+$ gradients and strongly suggests that the TOP-80S is not engaged in translating the mRNA (Infante and Baierlein, 1971; Beller and Lubsen, 1972; Noll et al, 1973) (see Appendix text on "TOP-80S shift in high-K$^+$ gradients"). Importantly, LARP1 protein undergoes the same redistribution as TOPs in high-K$^+$ gradients (Fig. 2B). Based on these data, we propose that the TOP-80S contains weakly-associated 40S and 60S subunits which are somehow complexed with LARP1 and with the TOP but are not directly engaged in translating the mRNA.

To directly evaluate the presence of 40S and 60S subunits in the salt-sensitive TOP-80S, we treated cells with Torin1 following siRNA-mediated knockdown of RPS24 (a 40S protein) or RPL31 (a 60S protein). These knockdowns reduced the overall abundance of their corresponding ribosomal subunit (and led to an accumulation of the opposite subunit) as previously observed for other ribosomal protein knockdowns (Luan et al, 2022; Gentilella et al, 2017). Importantly, si-RPS24 shifted TOPs from the TOP-80S to the free mRNA fraction (with no ribosomal subunits bound), while si-RPL31 shifted TOPs from the TOP-80S to the TOP-40S (Fig. 2C; Appendix Fig. S2D). In contrast, non-TOP migration patterns were mostly unaffected by these knockdowns (Appendix Fig. S2E). These data provide further evidence that the TOP-80S contains both a 40S and a 60S subunit. Moreover, these data demonstrate that the 40S joins the complex first (and directly) because the knockdown of the 40S protein prevents both subunits from joining, while the knockdown of the 60S protein prevents only 60S joining.

As a final test of the model that the TOP-80S does not represent actively translating ribosomes, we knocked down eIF5B, an initiation factor which mediates the late initiation step of 60S joining at AUG start codons (Lee et al, 2002). In untreated cells, si-eIF5B caused an increase in the monosome peak, confirming that the knockdown was sufficient to globally inhibit translation initiation (Appendix Fig. S2F). Strikingly, eIF5B knockdown had no impact on the migration or abundance of the TOP-80S under Torin1 treatment (Fig. 2C; Appendix Fig. S2D), demonstrating that 60S joining to the TOP-80S is not mediated by canonical mechanisms at an AUG start codon. Taken together, these data demonstrate that the TOP-80S contains a TOP complexed with LARP1 and weakly-associated 40S and 60S subunits that are not actively translating.

## An increase in free ribosomes is sufficient to drive TOP-80S formation

Having established key components comprising the TOP-80S, we were next interested in the molecular determinants that drive its formation. While mTOR signaling is considered a major determinant of TOP regulation (Fonseca et al, 2015; Hong et al, 2017; Philippe et al, 2018; Jia et al, 2021; Fuentes et al, 2021), we wondered whether other treatments resulting in an increase in free ribosomes would be sufficient to induce the formation of the TOP-80S.

As expected, siRNA-mediated knockdown of the initiation factors eIF4E, eIF4G, eIF4A1/2, eIF3B, or eIF2S1 each led to a global decrease in translation initiation and an increase in free ribosomes, evidenced by an increase in the monosome:polysome ratio (Fig. 3A,B). Importantly, however, none of these conditions led to a strong decrease in eIF4E-binding protein 1 (4EBP1) phosphorylation, a canonical readout of mTOR signaling (Thoreen et al, 2009) (Fig. 3A,B). Nonetheless, all conditions led to a redistribution of TOPs, but not non-TOPs, to the 80S fraction (TOP-80S) (Fig. 3A,B; Appendix Fig. S3A–D). We also pharmacologically increased free ribosomes by treating cells with Torin1 (positive control), silvestrol (eIF4A inhibitor (Wolfe et al, 2014)), sodium arsenite (inducer of eIF2α-phosphorylation and the integrated stress response (Duncan and Hershey, 1987; Pakos-Zebrucka et al, 2016)), or puromycin (tyrosyl-tRNA mimetic which releases nascent peptides and triggers dissociation of ribosome

subunits from mRNAs (Azzam and Algranati, 1973)); each treatment led to an increase in monosomes relative to polysomes (Fig. 3C). Treatment with each agent also led to the robust formation of the TOP-80S while only Torin1 caused an observable change in mTOR signaling (Fig. 3C; Appendix Fig. S3E,F). Collectively, these data establish that increases in free ribosomes promote the formation of the TOP-80S even without changes in mTOR signaling.

To formally establish the identity of the TOP-80S in these conditions, we repeated these experiments in LARP1-KO cells and in spread, high-K+ gradients. In all conditions, the TOP-80S failed to form in LARP1-KO cells (Appendix Fig. S4A–C) and sedimented distinctly upstream of the canonical 80S in high-K+ gradients (Appendix Fig. S4D,E). Together, these data establish that the TOP-80S complexes formed under these conditions correspond to the same LARP1-dependent, weakly-associated 40S and 60S subunit complexes described earlier.

Prevailing models implicate LARP1 phosphorylation at multiple residues as a key mediator of TOP regulation (Yu et al, 2011; Kang et al, 2013; Hong et al, 2017; Philippe et al, 2018; Fonseca et al, 2018; Jia et al, 2021). Given that none of the perturbations impacted mTOR signaling to 4EBP1, we wondered whether there were changes in LARP1 phosphorylation status downstream of these same stimuli. We observed that only Torin1 induced an obvious downshift in the migration pattern of LARP1 by Phos-tag gel while the other treatments did not (Fig. 3D; Appendix Fig. S4F). While we cannot exclude the possibility that small-scale phosphorylation changes to LARP1 mediate TOP-80S formation, these data show that global phosphorylation of LARP1 is not appreciably decreased in these regimes, and that formation of TOP-80S does not require dephosphorylation of LARP1.

## Cryo-EM structure reveals LARP1 bound in the mRNA channel of the 40S subunit

Intrigued by the association of LARP1 and TOPs with ribosomes, we solved a single-particle cryo-EM structure of mature human 40S ribosome complexes with LARP1 at 3.2 Å resolution, obtained from an in vivo immunoprecipitation using PYM1 (a known ribosome-associated factor (Diem et al, 2007)) as purification bait (Fig. 4A,B; Appendix Figs. S5A, S6A–F, S7A; Appendix Table S1). The resolved density maps allowed us to construct a de novo model of the 40S ribosome and unambiguously assign and fit AlphaFold (Jumper et al, 2021) predicted models for amino acids 660–724 of the long-isoform annotation of LARP1 (ENSEMBL LARP1-204) (Schwenzer et al, 2021). This assignment of LARP1 was confirmed by a second cryo-EM structure which we obtained using LARP1 as purification bait (Fig. 4C; Appendix Figs. S5B, S6A–E, S7B). Based on these two structures we were able to define a previously uncharacterized domain of LARP1 located between the La/PAM2 and DM15 domains, which we call the Ribosome Binding Region (RBR; Fig. 4A). According to the PDBePISA server (Krissinel and Henrick, 2007), the interaction surface area between the RBR and the 40S subunit is approximately 2673 square angstroms. While LARP1 is found in an identical position in both structures, the superior structure obtained from the PYM1 sample is used for the remainder of this work.

Only the RBR of LARP1 could be observed in our structure. It extends from the inter-subunit side to the solvent side of the 40S

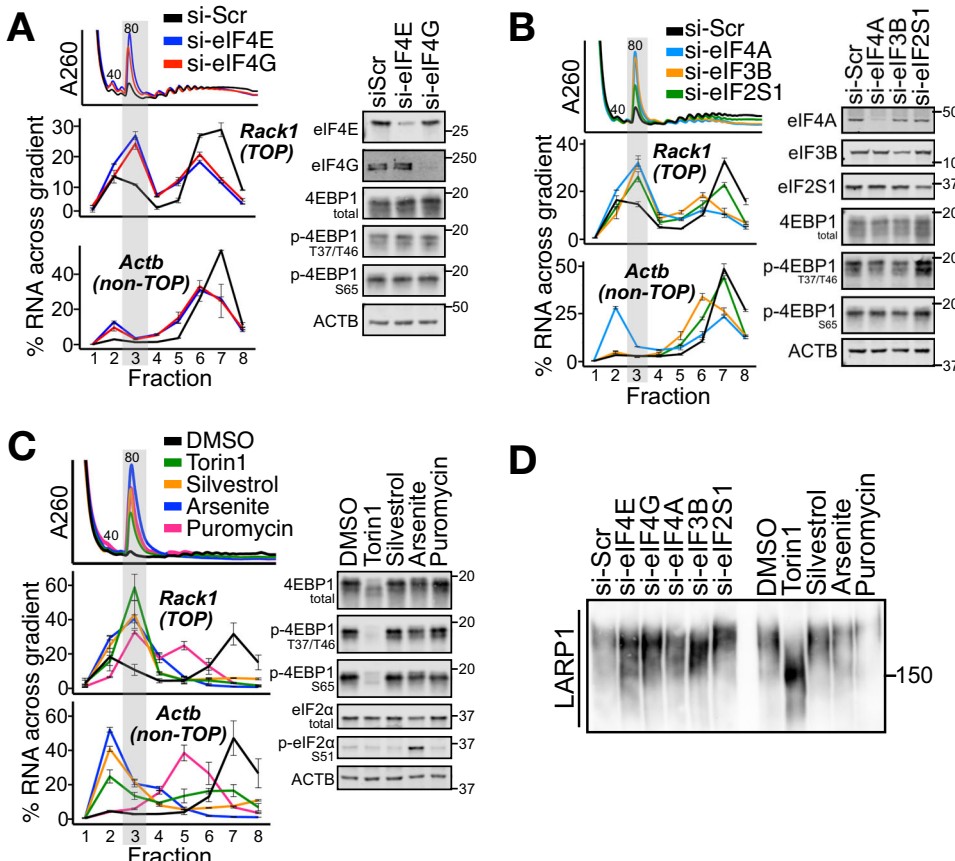

**Figure 3. Increases in free ribosomes drive TOP-80S formation.**

(A) HEK293T cells were treated with siRNAs targeting scrambled (si-Scr), eIF4E, or eIF4G and lysates fractionated along 10–50% sucrose gradients followed by qPCR against genes of interest. Western blots for knockdown efficiency and proteins of interest are shown. (B) Identical to (A) except with siRNAs targeting scrambled (si-Scr), eIF4A1/2, eIF3B, or eIF2S1. (C) Identical to (A) except cells were treated with Torin1 (300 nM, 1 h), Silvestrol (30 nM, 1 h), Sodium arsenite (Arsenite; 100 µM, 1 h), or Puromycin (250 µM, 30 min). (D) Phos-tag gel for LARP1 following identical treatments to those described throughout Fig. 3. For (A–D), gray highlights correspond to the TOP-80S; biological replicate experiments are shown in Appendix Figs. S3B, D, F, S4F, respectively. For qPCR plots, error bars are centered at the average and represent the SD of two to four technical replicates. Source data are available online for this figure.

subunit through the mRNA entry channel (Fig. 4B,C; Appendix Figs. S7A,B). The RBR can be subdivided into two segments of defined density separated by short stretches of unresolved density (Fig. 4D). The first segment of residues 660-694 traverses from the decoding center (DC) through the mRNA channel (Fig. 4D; Appendix Fig. S7C). The N-terminus of this segment is located within the DC where residue H665 stacks with base C1698 (C1397 in *E. coli*) of the 18S rRNA, which plays an important role during tRNA decoding (Fig. 4E) (Jenner et al, 2010). This localization agrees with CLIP-seq data identifying contacts between LARP1 and nucleotides 1698–1702 of the 18S rRNA (Wolin et al, 2023). The subsequent protein helix (residues 667–694) passes through the mRNA channel using basic residues to interact with 18S rRNA helices 1 and 18 (e.g., R668 with h1 and K670 with h18; Fig. 4F). Deeper in the channel the same helix makes hydrophobic contacts with ribosomal proteins RPS3, RPS2, and RPS30: Y685 of LARP1 stacks with R117 and R124 of RPS3 (Fig. 4G); Y686, L683 and I679 of LARP1 contact the hydrophobic surface of RPS2 (Fig. 4H); and W691 of LARP1 stacks with F49 of RPS30 (Fig. 4I). Following this helix, LARP1 becomes disordered and the second

segment of the RBR emerges and includes residues 708–724 which make hydrophobic contacts with the C-terminal tail of RPS3 and with the N-terminal domain of RPS17 (Fig. 4J; Appendix Fig. S7D). Finally, an unknown density is observed on the surface of RACK1 which we were unable to confidently assign due to the lack of structural features and low local resolution (Appendix Fig. S7E). This density could belong to LARP1 or to a different protein.

The cryo-EM structure of LARP1 bound to the 40S subunit within the DC and mRNA channel suggests that the complex is translationally inactive. Indeed, superimposing the mRNA density from a solved structure of a scanning 48S preinitiation complex (PIC) (Brito Querido et al, 2020) reveals that the LARP1 RBR directly clashes with mRNA density in the channel and would occlude mRNA binding (Appendix Fig. S8A). We further speculate that LARP1 binding to the DC could hinder the association of eIF1A, a critical initiation factor (Appendix Fig. S8B) (Yi et al, 2022). Finally, the LARP1 RBR occupies a similar position in the mRNA channel to known translation repressors SARS-CoV-2 NSP1 (Thoms et al, 2020) (Appendix Fig. S8C) and human SERBP1 (Wells et al, 2020) (Appendix Fig. S8D), confirming that the

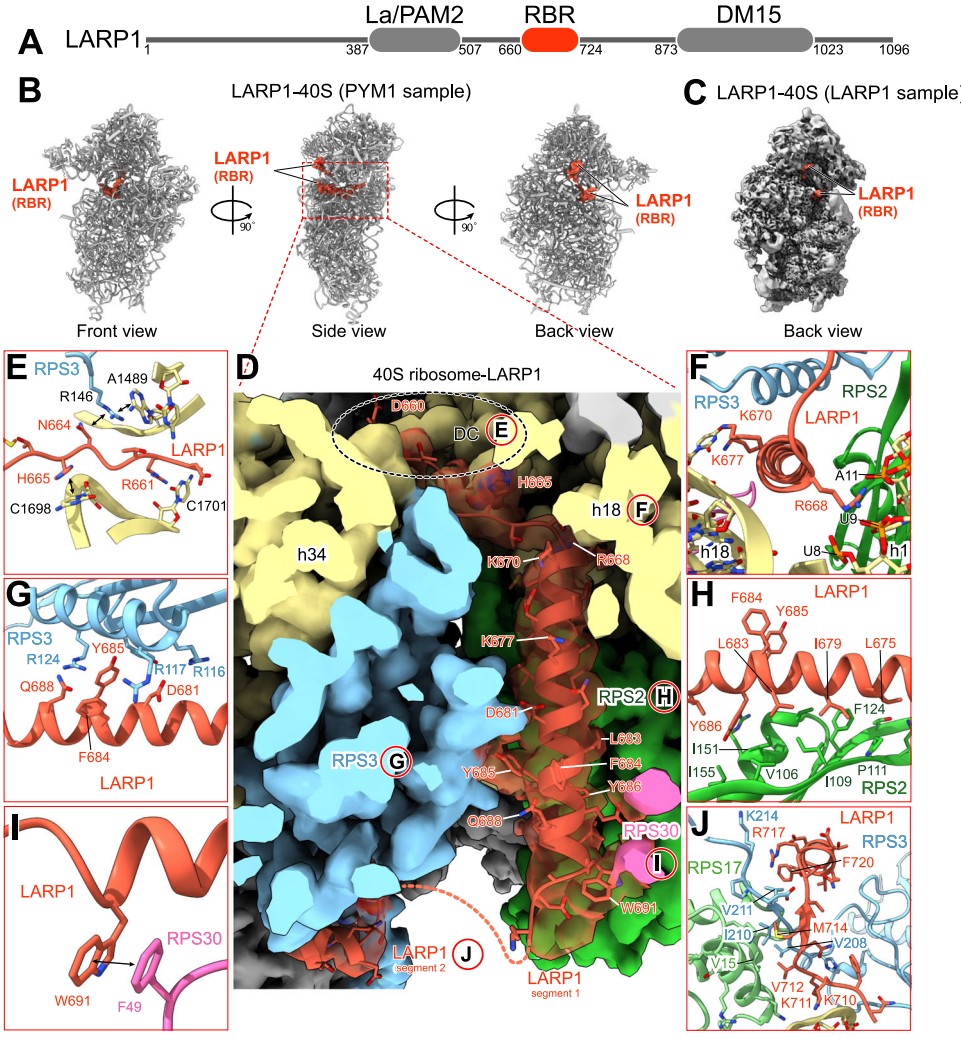

**Figure 4. Cryo-EM analysis of the interaction between LARP1 and the 40S ribosome.**

(A) Schematic showing the domain arrangement of LARP1 protein. (B) Three different views of the cryo-EM structure of the human LARP1-40S ribosome complex (from the PYM1 sample). (C) Back view of cryo-EM structure of the LARP1-40S ribosome complex (from the LARP1 sample). (D) Zoom-in of the RBR of LARP1 bound within the mRNA channel of the 40S ribosomal subunit. References to specific figure panels are annotated with the corresponding letter circled in red. (E) LARP1 residues 660–665 bind to the decoding center of the 40S ribosome. Key residues interacting with RPS3 (blue) and 18S rRNA (yellow) are shown. (F–I) Detailed schematics of LARP1 residues 666–694; key residues interacting with 18S rRNA helices 1 and 18 (yellow, F), RPS3 (blue, G), RPS2 (green, H) and RPS30 (pink, I) are shown. (J) LARP1 residues 708–724 interacting with 18S rRNA (yellow), RPS3 (blue), and RPS17 (light green) are shown. For (D–J), residues and nucleobases are annotated, and stacking interactions are indicated by black bidirectional arrows.

LARP1-40S complex is translationally inactive. Importantly, while we were unable to solve a structure of LARP1-80S complexes, the position of the RBR in the mRNA channel would be compatible with 60S subunit joining to form LARP1-80S complexes (Appendix Fig. S8E).

## LARP1 simultaneously binds ribosomes and TOPs via multimodal interactions

To validate the cryo-EM structure, we introduced nine alanine point mutations into LARP1 at contacts with the 40S subunit ("LARP1-RBRmut") (Fig. 5A). We transfected plasmids expressing either wild-type (WT) LARP1 or RBRmut-LARP1 (behind a partial CMV promoter) into LARP1-KO cells and treated with Torin1 to

evaluate formation of the TOP-80S. Importantly, WT and RBRmut-LARP1 expressed equally from these plasmids and their levels were similar to endogenous levels (Fig. 5B; Appendix Fig. S9A). We note that these constructs express the short isoform of LARP1 (LARP1-201), which is slightly lower molecular weight (but functionally equivalent) to the longer isoform expressed in HEK293T cells (Schwenzer et al, 2021; see Appendix Text on "LARP1 isoforms"). While the TOP-80S forms in cells expressing WT-LARP1, this complex fails to form in cells expressing RBRmut-LARP1 (Fig. 5C; Appendix Fig. S9B). By comparison, the sedimentation profile of a non-TOP, Ubl5, is largely unperturbed in this background (Fig. 5C; Appendix Fig. S9B). These data establish that the RBR of LARP1 is required for the formation of the TOP-80S. Furthermore, while WT-LARP1 protein

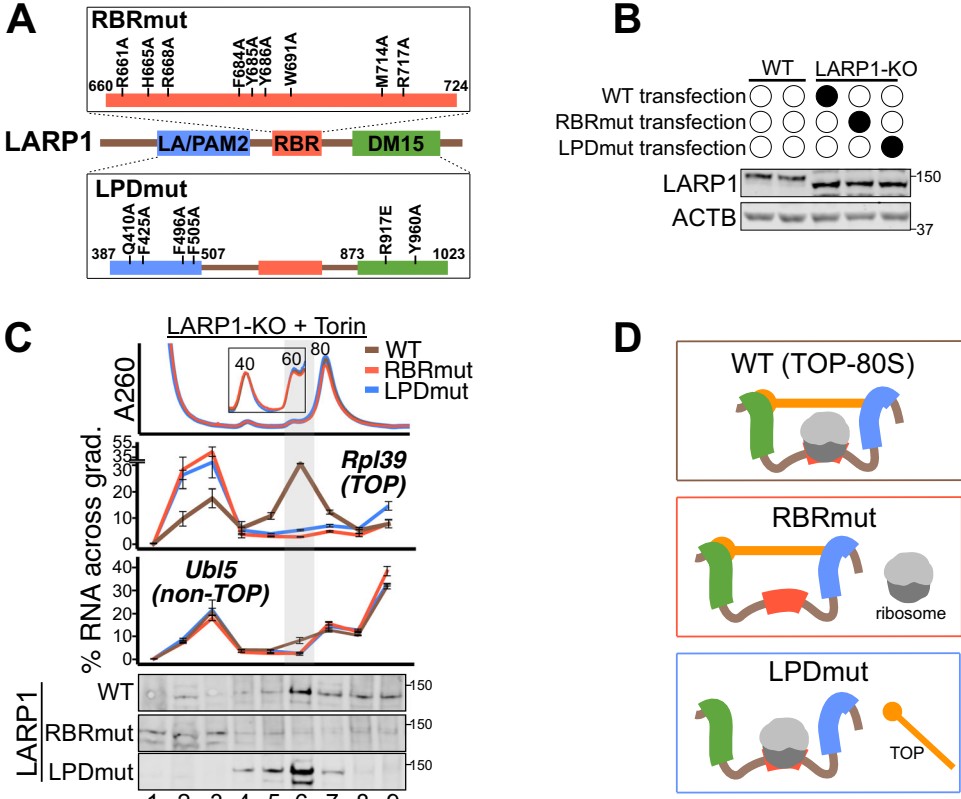

**Figure 5. LARP1 simultaneously binds ribosomes and TOPs via multimodal interactions.**

(A) Schematic showing the amino acid substitutions for the RBRmut- and LPDmut-LARP1 constructs. Amino acid numbering corresponds to the ENSEMBL LARP1-204 ("long isoform") annotation. (B) Western blot for LARP1 levels from WT or LARP1-KO cells transfected with the indicated constructs. Two independent samples are shown for WT cells (lanes 1 and 2). (C) LARP1-KO cells were transfected with the indicated constructs, followed by treatment with 300 nM Torin1 for 1 h. Lysates were fractionated along 15–35% high-K$^+$ (200 mM KOAc) sucrose gradients followed by qPCR against genes of interest. Error bars are centered at the average and represent the SD of two to four technical replicates. Gray highlight corresponds to the TOP-80S. Western blots against LARP1 from the same samples are presented below qPCR traces. (D) Schematic of the complexes formed by the indicated LARP1 proteins. TOP mRNA (orange), ribosomes (gray), LARP1 (brown), La/PAM2 (blue), RBR (red), and DM15 (green) domains are depicted. WT-LARP1 binds both ribosomes and TOPs to form the TOP-80S. RBRmut-LARP1 fails to bind ribosomes. LPDmut-LARP1 fails to bind TOPs. For (B, C), biological replicate experiments for RBRmut-LARP1 are shown in Appendix Fig. S9A, B. Source data are available online for this figure.

preferentially sediments with the weakly-associated 80S fraction, the RBRmut protein does not (Fig. 5C). These data confirm that RBRmut-LARP1 cannot form the TOP-80S due to its inability to bind ribosomes.

We also introduced point mutations to the mRNA-binding La, PAM2, and DM15 domains of LARP1 (termed "LPDmut") (Fig. 5A); these point mutations prevent binding to polyA, PABPC1, and the TOP motif, respectively (Mattijssen et al, 2021; Kozlov et al, 2022; Lahr et al, 2017), and are therefore predicted to preclude mRNA binding interactions by LARP1. LPDmut-LARP1 is expressed at equivalent levels relative to WT (Fig. 5B), and the protein maintains its sedimentation with the weakly-associated 80S fraction in high-K$^+$ gradients (Fig. 5C; bottom); however, despite its strong association with ribosomes, LPDmut-LARP1 does not recruit the TOP into the weakly-associated 80S fraction (Fig. 5C; top). Thus, LPDmut-LARP1 fails to form the TOP-80S due to an inability to engage the TOP mRNA. These data establish that the RBR can bind ribosomes in the absence of known mRNA binding interactions by LARP1. Collectively, these data establish that the RBR of LARP1 directly binds ribosomal subunits while the La, PAM2,

PAM2, and/or DM15 domains simultaneously bind the TOP, together forming the TOP-40S and TOP-80S complexes (Fig. 5D).

## Ribosome binding is not required for LARP1-mediated TOP repression or stabilization

Having established the molecular characteristics of LARP1-ribosome binding and its implications for the formation of the TOP-40S and TOP-80S complexes, we next interrogated the functional outcomes of this binding event. LARP1 has been shown in multiple studies to translationally repress and stabilize TOPs (Philippe et al, 2018; Fonseca et al, 2015; Gentilella et al, 2017; Hochstoeger et al, 2024; Aoki et al, 2013; Philippe et al, 2020; Fuentes et al, 2021; Schneider et al, 2022). We, therefore, predicted that ribosome binding by LARP1 would serve as a sensor to drive TOP repression and stabilization when free ribosomes are abundant, thereby reducing the production of new ribosomal proteins (and new ribosomes).

To probe translational repression, we constructed rapidly degraded nano-luciferase reporters (nLuc-PEST) driven by the promoter and

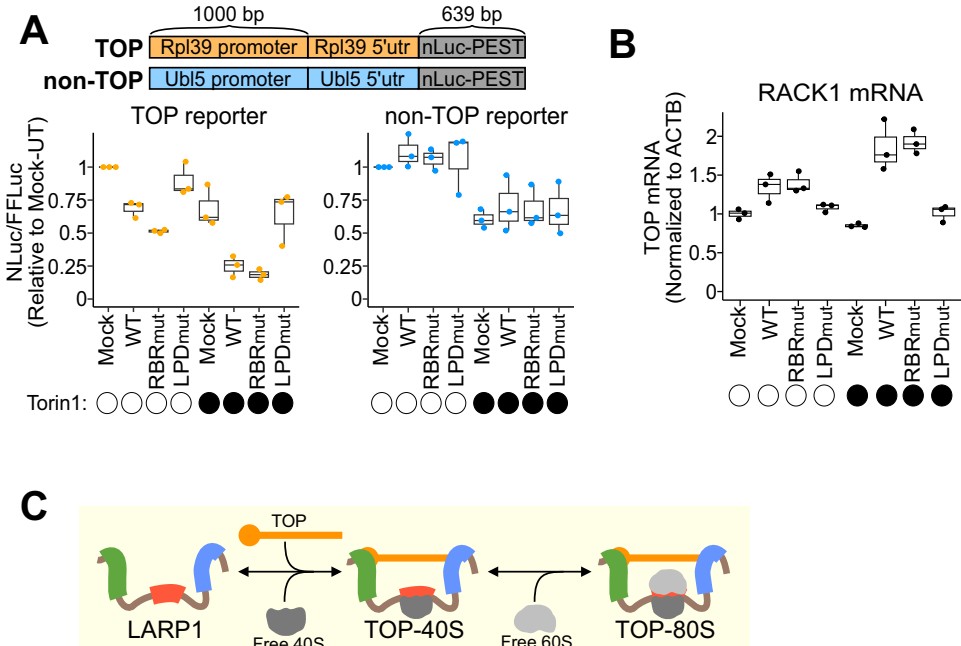

**Figure 6.  LARP1-ribosome binding is not required for TOP repression or stabilization.**

(**A**) Top: Schematic showing the reporter constructs. FFluc expressed from a CMV promoter on the same plasmid served as a transfection control. Bottom: NLuc-PEST expression (normalized to FFluc) from three biological replicates for the indicated reporters and conditions are shown. Values were normalized to the mock-transfected, untreated sample. Torin1 treatment was 300 nM for 90 min. (**B**) Steady-state mRNA levels (qPCR) from three biological replicates for RACK1 normalized to ACTB mRNA. Values were normalized to the mock-transfected, untreated sample. Torin1 treatment was 300 nM for 24 h. (**C**) Model figure: Repressed LARP1-TOP complexes form by sequential binding of the 40S and 60S subunits to the RBR of LARP1 while the DM15 and La/PAM2 domains bind the TOP. TOP mRNA (orange), ribosomal subunits (gray), La/PAM2 (blue), RBR (red), and DM15 (green) domains of LARP1 (brown) are depicted. For box plots in (A-B): whiskers represent minima and maxima; bounds of box represent the quartiles (25th and 75th percentile); center line represents the median. Source data are available online for this figure.

5'UTR of either Rpl39 (TOP) or Ubl5 (non-TOP) (Fig. 6A). Firefly luciferase (FFluc) was expressed from a CMV promoter on the same plasmid to serve as a transfection control. As expected, nLuc-PEST expression from the TOP reporter was more sensitive to Torin1 treatment than the non-TOP reporter (Appendix Fig. S10A). We also observed robust translation repression of the TOP reporter in response to silvestrol or puromycin, but it was not more sensitive than the non-TOP reporter to these perturbations (Appendix Fig. S10A). These data suggest that increased free ribosomal subunits are alone insufficient to drive preferential TOP repression (over non-TOPs). We next transfected these validated reporters into LARP1-KO cells co-transfected with mock, WT-, RBRmut-, or LPDmut-LARP1 constructs. As expected, WT-LARP1 repressed the TOP reporter and this effect was enhanced by Torin1 treatment (Fig. 6A). Surprisingly, cells expressing RBRmut-LARP1 displayed equal or greater TOP repression compared to WT-LARP1 both in untreated and Torin1 conditions (Fig. 6A). LPDmut-LARP1, in contrast, was incapable of repressing the TOP reporter both in untreated and Torin1 conditions (Fig. 6A), consistent with its inability to bind the mRNA. As a control, all LARP1 constructs minimally affected nLuc-PEST expression from the non-TOP reporter (Fig. 6A). Taken together, these data establish that, contrary to our initial expectations, ribosome binding by LARP1 does not drive TOP repression.

We next evaluated TOP stability by transfecting LARP1-KO cells with mock, WT-, RBRmut-, or LPDmut-LARP1 and performing qPCR for endogenous TOP levels. Consistent with

earlier studies (Gentilella et al, 2017; Fuentes et al, 2021; Schneider et al, 2022; Hochstoeger et al, 2024; Aoki et al, 2013), the presence of WT-LARP1 increased TOP steady-state levels and this effect was enhanced after 24 h of Torin1 (Fig. 6B; Appendix Fig. S10B); however, there was again no clear difference between the WT- or RBRmut-LARP1 constructs. In contrast, LPDmut-LARP1 was not able to confer an increase in TOP steady-state levels in either untreated or Torin1 conditions (Fig. 6B; Appendix Fig. S10B). These data indicate that ribosome binding by LARP1 is not required for TOP stabilization.

Collectively, these data argue that the association of ribosomal subunits with the TOP-40S and TOP-80S complexes does not play an essential role in two canonical LARP1 functions—TOP repression or stabilization.

## Discussion

Here we have identified a new domain of LARP1, the ribosome binding region (RBR), which directly interacts with 40S ribosomal subunits in a manner that prevents mRNA binding. Through this discovery and accompanying biochemical evidence, we have defined the molecular identities of the perplexing TOP-40S and TOP-80S complexes (Gentilella et al, 2017; Hong et al, 2017; Schneider et al, 2022; Fuentes et al, 2021). Our data indicate that the TOP-40S/80S form via binding of LARP1's RBR to a

translationally inactive 40S subunit while its La/PAM2 and DM15 domains bind the TOP, and the 60S joins at the RBR-40S interface through subunit interactions that are salt-sensitive (Fig. 6C). While we observe that formation of these complexes is driven by the availability of free ribosomal subunits and independent of observable phosphorylation changes in 4EBP1 and LARP1, we acknowledge that signaling downstream of mTOR likely provides additional critical layers of regulation (Fonseca et al, 2015; Hong et al, 2017; Philippe et al, 2018; Fonseca et al, 2018; Jia et al, 2021). Furthermore, 4EBP1 phosphorylation will certainly be involved in governing the access of LARP1's DM15 domain to the cap structure of TOPs to form these repressed complexes (Thoreen et al, 2012; Hochstoeger et al, 2024; Lahr et al, 2017).

While our data provide insight into the molecular nature of the TOP-40S/80S complexes (Gentilella et al, 2017; Fuentes et al, 2021; Schneider et al, 2022), we were surprised to find that ribosome binding to LARP1 does not mediate TOP repression or steady-state levels. This would have been an elegant mechanism to sense free ribosomal subunits and tune TOP activity and expression accordingly. Intriguingly, previous studies have attributed changes in TOP levels and stability to direct protection conferred by 40S subunits within the TOP-40S complex (Gentilella et al, 2017; Fuentes et al, 2021). Our cryo-EM structure provided a means to mutate residues involved in this interaction. In doing so, we demonstrate that TOP levels are established by LARP1 (and increased by mTOR inhibition) independent of 40S subunit binding within these complexes. In reconciling these seemingly contradictory pieces of evidence, we note that the ribosomal subunit depletions used to establish the prior model (Gentilella et al, 2017; Fuentes et al, 2021) cause widespread effects on cell physiology (Luan et al, 2022; Cheng et al, 2019) and therefore do not directly implicate 40S subunit binding within the TOP-40S as the driver of TOP stabilization.

In our reporter assay we observed somewhat greater TOP repression with the RBRmut construct compared to WT. These data are consistent with a previous study showing constitutive TOP repression from a truncated LARP1 construct which lacks the RBR (Philippe et al, 2018). Instead of promoting TOP repression, these data suggest that ribosome binding could instead promote TOP translation in a manner which is presently unclear. Perhaps the bound ribosomal subunits somehow function in the re-initiation of TOP translation following mTOR reactivation, as has been previously proposed (Fuentes et al, 2021; Ogami et al, 2022; Schneider et al, 2022). Another possibility is that LARP1-ribosome binding functions solely to limit the activity of bound ribosomal subunits. However, LARP1 exists at an abundance of ~$10^5$ copies per mammalian cell (Wiśniewski et al, 2014), roughly stoichiometric with mRNA (Velculescu et al, 1999) but at least an order of magnitude less abundant than ribosomes (Duncan and Hershey, 1983). As such, LARP1-ribosome binding would seem better suited to the regulation of TOPs than to the sequestration of inactive ribosomes. Finally, LARP1 protein levels are highly correlated with ribosomal protein levels across diverse cell types (Hong et al, 2017), but the mechanistic underpinnings of this correlation are not understood. It is plausible that ribosome binding by LARP1 could be involved in this co-regulation of ribosome and LARP1 levels.

Taken together, our data stimulate a rethinking of how ribosomal subunit interactions within the TOP-40S and TOP-80S regulate TOP outcomes. Considering the varying demands on ribosome numbers during cell differentiation (Saba et al, 2021), changes in cell size (Delarue et al, 2018; Fingar et al, 2002), and disease (Khajuria et al, 2018; Mills et al, 2017), we speculate that LARP1-ribosome binding could serve a general role in tuning TOP expression to the availability of free ribosomes across a diverse array of healthy and diseased mammalian cell contexts.

## Methods

**Reagents and tools table**

| Reagent/resource | Reference or source | Identifier or catalog number |
| --- | --- | --- |
| **Experimental models** | | |
| HEK293T cells | ATCC | CRL-3216 |
| U266B1 cells | ATCC | TIB-196 |
| HEK293/Flp-In/T-Rex | Invitrogen | R78007 |
| HEK293/Flp-In/T-Rex PYM1-StFLAG overexpression | This paper | N/a |
| HEK293/Flp-In/T-Rex LARP1-StFLAG overexpression | This paper | N/a |
| **Recombinant DNA** | | |
| pOG44 | Invitrogen | V600520 |
| pcDNA5-PYM1-StFLAG | This study | N/a |
| pcDNA5-LARP1-StFLAG | This study | N/a |
| pLJC1-LARP1-WT | Carson Thoreen (PMID 29244122) | N/a |
| pLJC1-LARP1-RBRmut | This study | N/a |
| pLJC1-LARP1-LPDmut | This study | N/a |
| Luciferase-pcDNA3 | Addgene | 18964 |
| TOP reporter (Rpl39-nLuc-PEST) | This study | N/a |
| Non-TOP reporter (Ubl5-nLuc-PEST) | This study | N/a |
| **Antibodies** | | |
| Rabbit anti-LARP1 | Cell Signaling | 70180 |
| Rabbit anti-eIF5B | Proteintech | 13527-1-AP |
| Rabbit anti-RPS24 | Abcam | ab196652 |
| Rabbit anti-RPL31 | Raybiotech | 144-64978 |
| Rabbit anti-ACTB | Cell Signaling | 4967S |
| Rabbit anti-EIF4E | Cell Signaling | 9742S |
| Rabbit anti-EIF4G | Cell Signaling | 2498S |
| Rabbit anti-4EBP1 | Cell Signaling | 9644S |
| Rabbit anti-P-4EBP1-T37-T46 | Cell Signaling | 2855S |
| Rabbit anti-P-4EBP1-S65 | Cell Signaling | 9451S |

| Reagent/resource | Reference or source | Identifier or catalog number |
|---|---|---|
| Rabbit anti-EIF2α/EIF2S1 | Cell Signaling | 9722S |
| Rabbit anti-P-EIF2α-S51 | Abcam | ab32157 |
| Rabbit anti-EIF4A | Cell Signaling | 2425S |
| Rabbit anti-EIF3B | Bethyl Laboratories | A301-761A |
| Goat anti-rabbit IgG secondary antibody | Licor | 926-32211 |
| Mouse anti-rabbit IgG-HRP secondary antibody | Santa Cruz Biotechnology | sc-2357 |
| **Oligonucleotides and other sequence-based reagents** | | |
| Non-targeting (scramble) siRNA | Horizon (Dharmacon) | D-001810-01-20 |
| RPS24 siRNA | Horizon (Dharmacon) | L-011155-00-0005 |
| RPL31 siRNA | Horizon (Dharmacon) | L-013587-00-0005 |
| EIF5B siRNA | Horizon (Dharmacon) | L-013331-01-0005 |
| EIF4E siRNA | Horizon (Dharmacon) | L-003884-00-0005 |
| EIF4G1 siRNA | Horizon (Dharmacon) | L-019474-00-0005 |
| EIF4A1 siRNA | Horizon (Dharmacon) | L-020178-00-0005 |
| EIF4A2 siRNA | Horizon (Dharmacon) | L-013758-01-0005 |
| EIF3B siRNA | Horizon (Dharmacon) | L-019196-00-0005 |
| EIF2S1 siRNA | Horizon (Dharmacon) | L-015389-01-0005 |
| qPCR primers | This study | Methods ("RT-qPCR" section) |
| **Chemicals, Enzymes and other reagents** | | |
| Hygromycin B | Thermo Fisher | 10687010 |
| Tetracycline | Thermo Fisher | A39246 |
| Protease inhibitor | Roche | 4693132001 |
| Cycloheximide | MilliporeSigma | C1988 |
| IGEPAL®CA-630 (NP-40) | MilliporeSigma | I8896 |
| FLAG Peptide | MilliporeSigma | F4799 |
| Anti-FLAG M2 Affinity Gel | MilliporeSigma | A2220 |
| DTT (Dithiothreitol) | Meilunbio | MB3047 |
| DMEM | Thermo Fisher | 11995073 |
| FBS | Thermo Fisher | A3160502 |
| Penicillin/Streptomycin | Thermo Fisher | 10378016 |
| Trypsin-EDTA | Thermo Fisher | 25200114 |
| RPMI 1640 GlutaMAX | Thermo Fisher | 61870127 |
| Sodium Pyruvate | Thermo Fisher | 11360070 |
| IL-6 recombinant protein | Thermo Fisher | PHC0066 |
| Lipofectamine RNAiMAX | Thermo Fisher | 13778150 |
| Lipofectamine 3000 | Thermo Fisher | L3000015 |

| Reagent/resource | Reference or source | Identifier or catalog number |
|---|---|---|
| OptiMEM | Thermo Fisher | 51985034 |
| Torin1 | Selleckchem | S2827 |
| Silvestrol | MedChemExpress | HY-13251 |
| Sodium Arsenite | MilliporeSigma | 106277 |
| Puromycin dihydrochloride | MilliporeSigma | P7255-100MG |
| PBS | Thermo Fisher | 11995065 |
| emetine dihydrochloride | MilliporeSigma | 324693 |
| NP-40 | MilliporeSigma | 492018 |
| TCEP | Gold-Bio | TCEP25 |
| SUPERaseIn | Thermo Fisher | AM2696 |
| Halt protease and phosphatase inhibitor | Thermo Fisher | 78444 |
| TURBO DNase | Thermo Fisher | AM2239 |
| Qubit RNA Broad Range assay kit | Thermo Fisher | Q10211 |
| RNaseA | Thermo Fisher | EN0531 |
| TRIzol | Invitrogen | 15596026 |
| eGFP mRNA | TriLink | L-7601-100 |
| GlycoBlue | Thermo Fisher | AM9516 |
| ProtoScript II | NEB | E6560L |
| SYBR Green | Bio-Rad | 1725124 |
| TCA | MilliporeSigma | T3699 |
| BCA Assay kit | Thermo Fisher | 23225 |
| 4–12% Criterion XT-Bis-Tris polyacrylamide gels | Bio-Rad | 3450125 |
| MES running buffer | Bio-Rad | 1610789 |
| Nitrocellulose membrane | Bio-Rad | 1704271 |
| PVDF membrane | Bio-Rad | 1704273 |
| Intercept Blocking Buffer | Licor | 927-60001 |
| Intercept Antibody Diluent | Licor | 927-65001 |
| Gibson Assembly Master Mix | NEB | E2611L |
| NanoGlo Dual-Luciferase Reporter Assay | Promega | N1630 |
| RIPA Lysis Buffer | Thermo Fisher | 89900 |
| Benzonase | MilliporeSigma | E1014 |
| Phos-Tag reagent | Wako | AAL-107 |
| EDTA-free pre-stained protein marker | Apex Bio | F4005 |
| Milk | Santa Cruz Biotechnology | sc-2325 |

| Reagent/resource | Reference or source | Identifier or catalog number |
|---|---|---|
| SuperSignal West Femto Maximum Sensitivity Substrate | Thermo Fisher | 34095 |
| **Software** | | |
| AlphaFold | Jumper et al | https://alphafold.com |
| ChimeraX | Goddard et al | https://www.cgl.ucsf.edu/chimerax/ |
| Coot v0.9.8.5 | Emsley et al | https://www2.mrc-lmb.cam.ac.uk/personal/pemsley/coot/ |
| CryoSPARC | Punjani et al | https://cryosparc.com/ |
| Phenix | Adams et al | https://phenix-online.org |
| MolProbity | Chen et al | http://molprobity.biochem.duke.edu/ |
| RefMac5 | Yamashita et al | https://www.ccp4.ac.uk/html/refmac5.html |
| 3DFSC | Tan et al | https://3dfsc.salk.edu |
| Guppy | Oxford Nanopore Technologies (ONT) | https://nanoporetech.com/document/Guppy-protocol |
| MinKNOW | Oxford Nanopore Technologies (ONT) | https://nanoporetech.com/document/experiment-companion-minknow |
| Minimap2 | PMID 29750242 | https://github.com/lh3/minimap2 |
| Samtools | PMID 19505943 | http://www.htslib.org |
| R (coding language) | Comprehensive R Archive Network | https://www.r-project.org |
| biomaRt | PMID 19617889 | https://bioconductor.org/packages/release/bioc/html/biomaRt.html |
| ggplot2 | Wickham H (2016). ggplot2: Elegant Graphics for Data Analysis. | https://ggplot2.tidyverse.org |
| **Other** | | |
| Quantifoil R1.2/1.3 Grids +2 nm C Layer | Quantifoil | 4220CC-XA |
| Chromatography column | Bio-Rad | 7326008 |
| Amicon Ultra-0.5 Centrifugal Filter Unit | Millipore | UFC5100BK |
| SW41 ultracentrifuge tube | Seton Scientific | 7030 |
| SW41 swinging bucket rotor | Beckman | N/a |
| RNA pico bioanalyzer chip | Agilent | 5067-1513 |
| Nanopore PCR-cDNA Barcoding kit | Oxford Nanopore Technologies (ONT) | SQK-PCB109 |
| Genomic DNA Screen Tape | Agilent | 5067-5366 |

| Reagent/resource | Reference or source | Identifier or catalog number |
|---|---|---|
| R9.4.1 MinION flowcells | Oxford Nanopore Technologies (ONT) | FLO-MIN106 |
| GridION Mk1 sequencer | Oxford Nanopore Technologies (ONT) | GRD-MK1 |
| QuantStudio 6 qPCR machine | Thermo Fisher | 4485691 |
| Odyssey CLx Imager | Licor | Model no. 9140 |
| TeloPrime Full Length cDNA Amplification Kit V2 | Lexogen | 013.08 |
| BioTek Synergy H1 Microplate Reader | Agilent | N/a |
| 10 cm dishes | Corning | CLS430167 |
| 6-well dish | Corning | 3516 |
| T225 flask | CytoOne | CC7682-4822 |
| 96-well TC-treated plates | Corning | 3603 |
| ChemiDoc imaging system | Bio-Rad | 12003153 |

## Cell culture

All HEK293T (ATCC CRL-3216) and HEK293/Flp-In/T-Rex (Invitrogen R78007) cell lines were cultured in DMEM (Thermo Fisher 11995073) supplemented with 10% FBS (Thermo Fisher A3160502). The HEK293/Flp-In/T-Rex were additionally cultured with 1x penicillin/streptomycin (Thermo Fisher 10378016). To begin experiments, cells were trypsinized in 0.25% Trypsin-EDTA (Thermo Fisher 25200114), pelleted at $350 \times g$ for 4 min, resuspended in DMEM/FBS, and seeded to the appropriate concentration in tissue culture dishes (Corning). Experiments were started the following day after allowing one overnight for cell attachment. U266B1 cells (ATCC) were cultured in RPMI 1640 GlutaMAX media (Thermo Fisher 61870127) supplemented with 20% FBS, 1 mM sodium pyruvate (Thermo Fisher 11360070), and 5 ng/uL human IL-6 recombinant protein (Thermo Fisher PHC0066).

## siRNA knockdowns

All siRNAs were purchased from Horizon Discovery as ON-TARGETplus SMARTPools according to the following catalog numbers:

Non-targeting (Scramble): D-001810-01-20
RPS24: L-011155-00-0005
RPL31: L-013587-00-0005
EIF5B: L-013331-01-0005
EIF4E: L-003884-00-0005
EIF4G1: L-019474-00-0005
EIF4A1: L-020178-00-0005
EIF4A2: L-013758-01-0005
EIF3B: L-019196-00-0005
EIF2S1: L-015389-01-0005

SMARTPool siRNAs were resuspended to 50 µM in siRNA buffer (10 mM Tris, 60 mM KCl, 0.2 mM MgCl$_2$) and stored as aliquots at −20 °C.

SiRNA knockdowns were performed for 48–96 h using Lipofectamine RNAiMAX transfection reagent (Thermo Fisher 13778150) according to the manufacturer's instructions with some modifications. Cells were seeded one day prior to initiating siRNA treatment and allowed 24 h to attach. For a 10 cm dish containing 10 mL media, 2–5 µL of 50 µM siRNA stock was resuspended in 500 µL OptiMEM reduced serum media (Thermo Fisher 51985034) and gently inverted to mix. Simultaneously, 30 µL of RNAiMAX transfection reagent was added to 500 µL OptiMEM and gently inverted to mix. After 5 min incubation at room temperature, the RNAiMAX-OptiMEM mixture was added dropwise to the siRNA-OptiMEM mixture and gently inverted to mix. After 10 min incubation at room temperature, the entire mixture was added dropwise to the cell media and gently swirled to mix (10–25 nM final siRNA concentration per well). The next day, the cell media was changed, and the same siRNA treatment was performed. In total, two siRNA transfections were performed for each dish. Cells were incubated for an additional 24–72 h following the second siRNA transfection before lysis.

SiRNA transfections in six-well dishes were performed as above except that reagent amounts were decreased to maintain the same concentration in 3–4 mL of media.

## Sucrose gradient fractionation

Cells were seeded at 0.6–3 × 10$^6$ cells per dish in 10 cm dishes (Corning CLS430167) and allowed to attach for ~24 h before starting on the appropriate drug or siRNA treatment. On the day of lysis, cells were replenished with fresh media 2 h prior to lysis. Cells were then started on appropriate drug treatments 20 min–1 h prior to lysis (Torin1 1 h prior; Silvestrol 1 h prior; Sodium Arsenite 1 h prior; Puromycin 20–30 min prior). At the time of lysis, cells were washed in 5 mL PBS (Thermo Fisher 11995065) containing 360 µM emetine dihydrochloride (MilliporeSigma 324693) and 200 µL of sucrose gradient lysis buffer was added dropwise to the plate (sucrose gradient lysis buffer recipe: 50 mM HEPES pH 7.4, 100 mM KOAc, 15 mM Mg(OAc)$_2$, 5% Glycerol, 0.25% (v/v) NP-40 Alternative (MilliporeSigma 492018), 360 µM emetine dihydrochloride, 1 mM TCEP (Gold-Bio TCEP25), 20 U/mL SUPERaseIn RNase Inhibitor (Thermo Fisher AM2696), 1x Halt protease and phosphatase inhibitor cocktail (Thermo Fisher 78444), 8 U/mL TURBO DNase (Thermo Fisher AM2239)). Cells were scraped directly from the plate in lysis buffer, gently pipetted to homogenize, transferred to ice for 10 min, and clarified by centrifuging at 8000 × g at 4 °C for 5 min.

Sucrose buffers were prepared on ice containing 25 mM HEPES pH 7.4, 100 mM KOAc, 5 mM Mg(OAc)$_2$, 180 µM emetine dihydrochloride, 1 mM TCEP, 4 U/mL SUPERaseIn RNase Inhibitor, and sucrose to the appropriate concentration (non-spread gradients: 10% and 50% (w/v) sucrose buffers; spread gradients: 15 and 35% (w/v) sucrose buffers). For high-K$^+$ gradients, the KOAc concentration in the sucrose buffers was 200 mM instead of 100 mM. Six mL of the low-percent sucrose buffer was pipetted to an SW41 ultracentrifuge tube (Seton Scientific 7030), after which 6 mL of the high-percent sucrose buffer was added to the bottom of the tube using a syringe and cannula. Sucrose gradients were mixed

on a Biocomp Gradient Master and placed at 4 °C until use on the same day.

Cell lysates were normalized to RNA content using the Qubit RNA Broad Range assay kit (Thermo Fisher Q10211), and equal amounts of RNA were layered on top of prepared sucrose gradients. For RNaseA treatments, 5 µg of RNaseA (Thermo EN0531) was added to 150 µg of RNA in a 200 µL reaction, shaken at 500 rpm for 20 min at 25 °C, and quenched with 200 U of SUPERaseIn on ice prior to layering on top of sucrose gradients. Gradients were ultracentrifuged in a Beckman SW41 swinging bucket rotor for either 75 min (non-spread gradients) or 300 min (spread gradients) at 40,000 rpm. Following ultracentrifugation, gradients were fractionated, fractions collected, and A260 absorbance measured using a Biocomp Piston Gradient Fractionator per the manufacturer's instructions. After fractionating, the volume in the bottom of the SW41 tube was also collected.

## RNA extractions

Whole-cell lysates or sucrose gradient fractions were taken directly to 3x volumes of TRIzol reagent (Invitrogen 15596026) supplemented with an equivalent amount of eGFP spike-in mRNA (TriLink Biotechnologies L-7601-100), vortexed, and frozen at −20 °C overnight. The eGFP mRNA spiked into each sample ranged from 30 pg to 1 ng, depending on the experiment. The next day, RNA was extracted from each sample according to the manufacturer's instructions with slight modifications. In brief, 0.2 mL of chloroform was added per 1 mL TRIzol reagent, and samples were vortexed and centrifuged at 16,000 × g at 4 °C for 15 min. After centrifuging, the top aqueous layer was taken to a new tube containing an equal volume of isopropanol and 1.5 µL GlycoBlue coprecipitant (Thermo Fisher AM9516). Samples were left at −20 °C for >1 h and centrifuged at 21,000 × g for 30 min. The pellet was washed once with 75% ethanol, spun at 21,000 × g for 10 min, supernatant aspirated, and air dried at room temperature. Once dry, the pellet was dissolved in 20–50 µL low TE buffer (10 mM Tris-HCl, 0.1 mM EDTA) and either used immediately or frozen at −80 °C.

## Nanopore sequencing across gradients

U266B1 cells were seeded at a density of 7.5 × 10$^5$ cells/mL in 55 mL of media (RPMI supplemented with 20% FBS, 1 mM sodium pyruvate, and 5 ng/µL IL-6) in a T225 flask (CytoOne CC7682-4822). The next day cells were treated with either 300 nM Torin1 or an equivalent volume of DMSO (vehicle) for 1 h. After 1 h, cells from two flasks per condition were pooled and spun at 1000 × g for 1 min at room temperature, the media aspirated, and the cell pellet was directly lysed in 150 µL of sucrose gradient lysis buffer. Lysates were gently pipetted to homogenize, transferred to ice for 10 min, and clarified by centrifuging at 8000 × g at 4 °C for 5 min. Cell lysates were normalized to RNA content using the Qubit RNA Broad Range assay kit (Thermo Fisher Q10211), and ~120 µg of RNA was sedimented along 10-50% sucrose gradients for 75 min and subsequently fractionated to 8 fractions. Thirty pg of eGFP mRNA was spiked into each fraction, and RNA was extracted as described in "RNA extractions." Five µg of extracted RNA from each fraction was subjected to polyA-enrichment using the NEBNext Poly(A) mRNA Magnetic Isolation Module (NEB E7490) according to the manufacturer's instructions. At this stage, an RNA pico bioanalyzer chip (Agilent 5067-1513) was performed,

and RNA was confirmed to be of high quality. One ng of polyA-enriched RNA from each sample was subjected to the Nanopore PCR-cDNA Barcoding kit (ONT SQK-PCB109) according to the manufacturer's instructions with 14 cycles of PCR amplification for all libraries. All libraries were run on a Genomic DNA Screen Tape (Agilent 5067-5366) to confirm that the libraries were of high quality. Barcoded DMSO and Torin1 libraries were separately pooled to 100 fmol in 11 μL of elution buffer, and each pool was sequenced on R9.4.1 MinION flowcells (ONT; FLO-MIN106) run on a GridION Mk1 sequencing device (ONT; GRD-MK1) for 72 h with an initial bias voltage of −180 mV. In total, nine barcoded libraries were sequenced on each flow cell (i.e., DMSO: 8 fractions and whole-cell library).

## Analysis of nanopore sequencing data

Sequencing was basecalled in real-time using the on-device Guppy basecaller (v4.2.3) in the MinKNOW software (v20.10.6) using the "high-accuracy" basecalling model. Multiplexed samples were demultiplexed in real-time, also using the on-device Guppy basecaller with the following key settings: "barcoding_kits = ["SQK-PCB109"], trim_barcodes = "off", require_barcodes_both_ends = "off", detect_mid_strand_barcodes = "off", min_score=60".

Each demultiplexed library was separated into its own fastq file using the grep command at the command line. The reference transcriptome FASTA was downloaded from Gencode (human release 38) using the wget command (http://ftp.ebi.ac.uk/pub/databases/gencode/Gencode_human/release_38/gencode.v38.transcripts.fa.gz) and the sequence for the eGFP spike-in (TriLink L-7601) was manually appended to the reference transcriptome (https://www.trilinkbiotech.com/media/folio3/productattachments/product_insert/egfp__orf_catno_l-7201_l-7601_l-7701_.txt).

Each library was then mapped to the transcriptome (using minimap2 with the -ax splice option), non-primary alignments excluded (using samtools view with the -F 256 option), and each output BAM file sorted (using samtools sort). Alignments were quantified by counting the number of reads aligning to each transcript within each BAM file (using the uniq -c command at the command line), and raw counts were output as a CSV file. CSV files were subsequently imported into R for further analysis and plotting.

In R, CSV files were joined into a large data frame. NA values were removed and transcripts were filtered for those with at least 80 total reads across all eight fractions combined (i.e., on average, 10 reads or more per fraction). Unmapped reads were removed, and transcriptome quantifications were normalized to the eGFP quantification within each sample. Percent RNA across the gradient was calculated as the eGFP-normalized value within each fraction relative to the total for all fractions across the gradient. To collapse transcriptome quantifications to the gene level, the ENSEMBL canonical transcript was then pulled from biomaRt (ENSEMBL version 110), and only the canonical transcript for each gene was kept in the data frame. Plots were constructed in R using the ggplot2 package. All relevant code is published on GitHub at repository 2023_Saba_Larp1 (https://github.com/jakesaba/2023_Saba_LARP1).

## RT-qPCR

Reverse transcription (RT) of extracted RNA (either from whole-cell lysates or sucrose gradient fractions) was performed using the ProtoScript II First Strand cDNA Synthesis Kit (NEB E6560L) according to the manufacturer's instructions with some modifications. Six μL of extracted RNA was added to 2 μL of random hexamer primer, heated to 95 °C for 2 min, and immediately placed on ice. To each reaction, 10 μL of 2x reaction mix and 2 μL of enzyme mix were added and mixed by pipetting on ice. Reactions (20 uL total) were incubated at 25 °C for 5 min, 42 °C for 60 min, and 80 °C for 5 min in a thermal cycler.

Following RT, cDNA was diluted 1:4 and 4.3 μL of cDNA was mixed with 5 μL iTaq Universal SYBR Green Supermix (Bio-Rad 1725124) and 0.7 μL of 5 μM forward and reverse qPCR primers targeting the gene of interest (333 nM final primer concentration) using the following cycle times: 95 °C for 2 min, [95 °C for 10 s, 60 °C 30 s] × 40 cycles, and then 60 °C for 3 min on a QuantStudio 6 RT-PCR system.

### Analysis

For each experiment, a standard curve containing seven 1:2 dilutions of a stock cDNA was run for each primer set, and the relative quantity of cDNA within each qPCR reaction was interpolated from the standard curve. Additionally, qPCR reactions targeting the eGFP spike-in RNA were performed for all sucrose gradient fractions within each experiment to control for differences in RNA extraction efficiency or RT. The quantification for each gene of interest was then normalized to that of the eGFP spike-in. Finally, the percent RNA across the gradient was calculated as a percentage of the eGFP-normalized cDNA value for each fraction relative to the total for all fractions across the gradient for that sample/qPCR target. RT-qPCR reactions were performed in duplicate or triplicate. Plots were constructed in R using the ggplot2 package.

All primer sets used in this study were validated by running an agarose gel of the product and observing a single band at the expected size. The primer sets are displayed below:

EGFP_F: CCCGACAACCACTACCTGAG
EGFP_R: GTCCATGCCGAGAGTGATCC
RACK1_F: GCTGATGGCCAGACTCTGTT
RACK1_R: TTCTAGCGTGTGCCAATGGT
RPL11_F: TCCATCATGGCGGATCAAGG
RPL11_R: TGTGCAGTGGACAGCAATCT
RPL5_F: CAGCGTATGCACACGAACTG
RPL5_R: ACCTATTGAGAAGCCTGCGG
RPL39_F: CTGCTCGCCATGTCTTCTCA
RPL39_R: CGAATCCACTGGGGAATGGG
RPS21_F: AATCGCATCATCGGTGCCAA
RPS21_R: CCTGTGACCTTGTCAACCTCG
ACTB_F: ACGTTGCTATCCAGGCTGTG
ACTB_R: GAGGGCATACCCCTCGTAGA
4EBP1_F: GGAGTGTCGGAACTCACCTG
4EBP1_R: ACTGTGACTCTTCACCGCC
ATF4_F: ATGGGTTCTCCAGCGACAAG
ATF4_R: GGAGAAGGCATCCTCCTTGC
NDUFB9_F: GCTTGTTTGATGAGAGCCC
NDUFB9_R: AGCACCATTCTGGGACCTTG
MYC_F: AGTGGAAAACCAGCAGCCTC
MYC_R: TTCTCCTCCTCGTCGCAGTA
UBL5_F: CTCGGGTGAGGAGCTGGT
UBL5_R: TCCTAGCTGGAGCTCGAATC
NDUFA1_F: GCGCATCTCTGGAGTTGATCG
NDUFA1_R: CAATGTTCTCCAAACCCTTTGACA

## TCA precipitation of proteins from sucrose gradients

Equal volumes of sucrose gradient fractions were resuspended in 15% trichloroacetic acid (TCA; MilliporeSigma T3699), mixed thoroughly, and stored at −20 °C overnight. The following day, samples were centrifuged at $21,000 \times g$ at 4 °C for 30 min, aspirated, and pellets washed 2x in 500 μL acetone (centrifuged at $21,000 \times g$ at 4 °C for 10 min and aspirated after each wash). After the final wash, pellets were dried at 42 °C in a vacuum evaporator for 5 min and resuspended in 2x Laemmli buffer. The total volume of resuspended protein was used for western blotting.

## Western blotting

For whole-cell lysates, clarified lysates were normalized to total protein (by BCA assay; Thermo Fisher 23225) or total RNA (by Qubit RNA Broad Range assay kit; Thermo Fisher Q10211) in 1X Laemmli buffer. Between 3 and 10 μg of protein or 1–5 μg of RNA was loaded per lane. For TCA-precipitated samples (from sucrose gradient fractions), the total amount of precipitated protein was loaded in each lane. Samples were boiled at 95 °C for 5 min and loaded into 4–12% Criterion XT-Bis-Tris polyacrylamide gels (Bio-Rad 3450125). Gel electrophoresis was performed in 1X MES running buffer (Bio-Rad 1610789) at 150 V for ~1 h. Gels were transferred to nitrocellulose membranes (Bio-Rad 1704271) and blocked in Intercept Blocking Buffer (Licor 927-60001) for 1 h at room temperature. Unless otherwise stated, samples were probed by primary antibody in Intercept Antibody Diluent (Licor 927-65001) at a 1:1000 dilution at 4 °C overnight. The next day, membranes were washed $3 \times 10$ min in TBST at room temperature and hybridized with 800CW goat anti-rabbit IgG secondary antibody (Licor 926-32211) at 1:5000 dilution for 1 h. After 1 h, membranes were washed $3 \times 10$ min in TBST at room temperature and washed $1 \times 10$ min in TBS. Blots were imaged using the Licor Odyssey CLx, and band intensities were quantified using Fiji.

All sources and catalog numbers for antibodies used in this study are provided below:

LARP1: Cell Signaling Technology 70180
EIF5B: Proteintech 13527-1-AP
RPS24: Abcam ab196652
RPL31: Raybiotech 144-64978
ACTB: Cell Signaling Technology 4967
EIF4E: Cell Signaling Technology 9742S
EIF4G: Cell Signaling Technology 2498S
4EBP1: Cell Signaling Technology 9644S
P-4EBP1_T37_T46: Cell Signaling Technology 2855S
P-4EBP1_S65: Cell Signaling Technology 9451S
EIF2α/EIF2S1: Cell Signaling Technology 9722S
P-EIF2α_S51: Abcam ab32157
EIF4A: Cell Signaling Technology 2425S
EIF3B: Bethyl Laboratories A301-761A

## LARP1-RBRmut and LARP1-LPDmut plasmid construction

pLJC1-LARP1 (WT-LARP1) expresses the C-terminally flag-tagged 1019 amino acid isoform of LARP1 (ENSEMBL LARP1-201; see Appendix text on "LARP1 isoforms") from a partial CMV promoter as previously described (Philippe et al, 2018). The plasmid was linearized by PCR, and gene blocks introducing

alanine point mutations at the appropriate sites were ordered from IDT and cloned into the linearized plasmid by Gibson assembly (NEB E2611L) according to the manufacturer's instructions. For the RBRmut construct, the point mutations correspond to the following sites of the 1096 amino acid LARP1 isoform (ENSEMBL LARP1-204) (Schwenzer et al, 2021): R661A, H665A, R668A, F684A, Y685A, Y686A, W691A, M714A, and R717A. For the LPDmut construct, the point mutations correspond to the following sites of LARP1-204: Q410A and F425A (La mutations (Kozlov et al, 2022)); F496A and F505A (PAM2 mutations (Mattijssen et al, 2021)); and R917E and Y960A (DM15 mutations (Lahr et al, 2017)) These LPD mutations are more commonly annotated in relation to the LARP1-201 isoform as the following: Q333A, F348A, F419A, F428A, R840E, and Y883A. All constructs were confirmed by whole-plasmid sequencing.

## Reporter construction and luciferase assays

Reporter constructs were generated in the Luciferase-pcDNA3 backbone (Addgene #18964). For the RPL39_nLuc-PEST construct, the plasmid was linearized, and its f1-ori, SV40 promoter, and Neo-resistance cassettes were excised by PCR. A geneblock containing the ENSEMBL RPL39-201 promoter (1000 bp), 5′UTR (54 bp), and an nLuc-PEST sequence was cloned into the linearized plasmid by Gibson Assembly. The resultant plasmid contained nLuc-PEST behind the RPL39 promoter/5′UTR and Firefly Luciferase (FFLuc) behind a CMV promoter. Extracts from HEK293T cells transfected with this reporter were subjected to the TeloPrime Full Length cDNA Amplification Kit V2 (Lexogen) followed by Nanopore sequencing, confirming the presence of the TOP motif at the 5′end of the transcribed mRNA. To construct the Ubl5_nLuc-PEST construct, the RPL39_nLuc-PEST plasmid was linearized, and the region corresponding to the RPL39 promoter and 5′UTR removed by PCR. A region corresponding to the ENSEMBL UBL5-202 promoter (1000 bp) and 5′UTR (420 bp) was amplified from HEK293T genomic DNA by PCR and cloned into the linearized plasmid by Gibson assembly. The resultant plasmid contained nLuc-PEST behind the UBL5 promoter/5′UTR and FFLuc behind a CMV promoter. All constructs were confirmed by whole-plasmid sequencing.

For luciferase assays using these constructs, cells were seeded between 5000 and 10,000 cells per well in 96-well TC-treated plates (Corning 3603). The next day, cells were transfected with 100 ng of the reporter plasmid using Lipofectamine 3000 according to the manufacturer's instructions. For co-transfection experiments, cells were co-transfected with 50 ng of the reporter plasmid and 150 ng of the appropriate LARP1 construct. The next day, cells were treated and subjected to the NanoGlo Dual-Luciferase Reporter Assay (Promega N1630) according to the manufacturer's instructions. Dual-luciferase readings were recorded using the Synergy H1 Microplate Reader (BioTek). NLuc measurements were normalized to the corresponding FFLuc measurement from each sample.

## Generation of tagged LARP1 and PYM1 cell lines

The human LARP1 and PYM1 genes were amplified from a cDNA library reverse transcribed from SK-HEP1 (ATCC HTB-52) cells and subcloned into a modified pcDNA5/FRT/TO plasmid (Invitrogen V652020), resulting in pcDNA5-LARP1-StFLAG and

pcDNA5-PYM1-StFLAG plasmids (C-terminal 2xStrep and 3xFLAG tag). The generation of the stable LARP1 and PYM1 cell lines were adapted from previous work (Ameismeier et al, 2018; Zemp et al, 2009). Briefly, HEK293/Flp-In/T-Rex cells were pre-cultured in a 10 cm dish for one day. At 50% confluence, the cells were co-transfected with 0.5 μg of the corresponding pcDNA5 plasmid and 4.5 μg recombinase plasmid POG44 (Invitrogen V600520). After 48 h, cells were passaged and subjected to a 14-day selection with 200 μg/ml hygromycin B (Thermo Fisher 10687010). The selected cell line was validated by Western blotting.

## Native complex preparation and LC-MS

Native ribosome-associated complexes were purified from a stable HEK293/Flp-In/T-Rex cell line as described previously (Ameismeier et al, 2018; Zemp et al, 2009). After two days of culture, expression of the bait protein (LARP1 or PYM1) was induced with 1 μg/ml tetracycline (Thermo Fisher A39246) for 24 h. Prior to collection, cells were treated with 10 μg/ml cycloheximide (CHX; MilliporeSigma C1988) for 10 min. A total of 50 dishes (15 cm) of cells were collected with a cell scraper, washed with cold 1X PBS buffer, and resuspended in lysis buffer (20 mM HEPES pH 7.4, 100 mM KOAc, 5 mM $MgCl_2$, 0.5% NP-40, 1 mM DTT, 10 μg/ml CHX, 0.5 mM NaF, 1 mM $Na_3V_3O_4$, 1X protease inhibitor mix (Roche 4693132001)). Cells were lysed by 10–15 strokes using a 15 mL douncer. The cell lysate was clarified by centrifugation at $10,000 \times g$ at 4 °C for 15 min, and the supernatant was incubated with 200 μL FLAG beads (Sigma A2220) in a 50 mL centrifuge tube for 2 h. After incubation, the beads were transferred to a small chromatography column, (Bio-Rad 7326008) washed once with lysis buffer, and then washed three times with wash buffer (20 mM HEPES pH 7.4, 100 mM KOAc, 5 mM $MgCl_2$, 10 μg/ml CHX). Finally, the protein complex was eluted with 500 μL elution buffer (0.4 mg/ml 3X Flag peptide (MilliporeSigma F4799) in wash buffer) for 45 min. The eluate was concentrated using a 100 kDa cut-off concentrator, and the final concentration was determined using a Nanodrop spectrophotometer. Protein quantification and identification were then performed by label-free protein quantification using LC-MS.

## Electron microscopy and image processing

Purified PYM1 or LARP1 samples (OD260 ≈ 5, ~0.1 μM) were stored in a buffer (20 mM HEPES pH 7.4, 100 mM KOAc, 5 mM $MgCl_2$, 10 μg/ml CHX, 0.4 mg/ml 3X Flag peptide, 0.05% NP-40). To prepare cryo-EM grids, commercially purchased R1.2/1.3 copper grids precoated with 2 nm continuous carbon-support (Quantifoil) were glow-discharge (PELCO easiGlow, TED PELLA) for 20 s at 15 mA with air. 3.5 μL of the samples were then applied to the grids and blotted for 4–5 s at 4 °C with the following settings 100% humidity, blot force 0, waiting time 60 s, and blot total 1. Subsequently, the grids were plunge-frozen in liquid ethane using an FEI Vitrobot Mark IV. The ø 55/20-mm blotting paper is made by TED PELLA and used for plunge freezing.

Data were collected on a Titan Krios G4 cryo-electron microscope operating at 300 keV using EPU 2. Cryo-EM data were collected with a pixel size of 1.146 Å/pixel (PYM1 sample) or 0.932 Å/pixel (LARP1 sample) and within a defocus range of −1 to −2.5 μm using a Falcon IV direct electron detector under low dose conditions with a total dose of 50 e-/Å² (PYM1 sample) or 58 e-/Å²

(LARP1 sample). Original image stacks were dose-weighted, aligned, summed, and drift-corrected using MotionCor2 (Zheng et al, 2017). Contrast transfer function (CTF) parameters and resolutions were estimated for each micrograph using GCTF (Zhang, 2016). Micrographs with an estimated resolution of less than 5 Å and astigmatism of less than 5% were manually inspected for contamination or carbon breakage.

### PYM1 sample

A total of 13,347 good micrographs were selected. Automatic particle picking was performed in Gautomatch without the use of a reference. Picked particles (1,906,164 particles) were extracted in Relion 3.1 (Zivanov et al, 2018) and then subjected to 2D classification in cryoSPARC (Punjani et al, 2017). In the end, 670,975 good particles clearly representing the 40S ribosome were selected for 3D classification in Relion 3.1 (Zivanov et al, 2018). After two rounds of alignment-free 3D classification, two classes – LARP1-40S ribosome and LARP1-LRRC47-40S ribosome—displaying strong LARP1 density, were selected and refined to high resolution in Relion (Zivanov et al, 2018). To obtain the final reconstruction, multibody refinement in Relion was used (Zivanov et al, 2018). The final maps were post-processed and local resolution filtered using masks that were automatically generated in Relion (Zivanov et al, 2018). Simultaneously, the final maps were sharpened using DeepEMhancer (Sanchez-Garcia et al, 2021). Detailed data processing procedures are shown in Appendix Fig. S5A.

### LARP1 sample

A total of 7643 good micrographs were selected. After automatic particle picking in Gautomatch, a total of 518,116 particles were extracted in Relion 3.1 and then subjected to 2D classification in cryoSPARC. Classes (107,728 particles) clearly representing the 40S ribosome were picked together. A non-uniform refinement (with default setting) was performed to obtain the final reconstruction. The final map was also local resolution filtered by cryoSPARC and DeepEMhancer. Detailed data processing procedures are shown in Appendix Fig S5B.

## Model building and refinement

In general, we used the human 80S-ribosome structures (PDB: 6Z6M) for rigid body fitting in the cryo-EM maps (Wells et al, 2020). Some manual adjustments were made in Coot (Emsley and Cowtan, 2004), particularly on the 40S head. The Alphafold (Jumper et al, 2021) predicted models of LARP1 and LRRC47 were rigid bodies fitted into their corresponding densities in Coot, then manually adjusted according to the high-resolution maps in Coot. Since the unknown density bound on top of RACK1 does not have enough resolution for unambiguous assignment, we did not build a model for it. All reported resolutions are based on the gold standard Fourier Shell Correlation (FSC) = 0.143 criterion.

The final models were real-space refined (default setting) with secondary structure restraints using the PHENIX suite (Adams et al, 2010). Final model evaluation was performed using MolProbity (Chen et al, 2010). We also used RefMac5 (Yamashita et al, 2021) to calculate the average FSC number between our maps and models, and used 3DFSC software (Tan et al, 2017) to calculate the directional FSC curves of our maps. Maps and models were visualized, and figures were generated using ChimeraX (Goddard et al, 2018).

## Phos-Tag gel immunoblotting

Cells were seeded to $1–3 \times 10^5$ cells per well in a six-well dish (Corning 3516) and allowed to attach for ~24 h before starting on the appropriate drug or siRNA treatment. On the day of lysis, cells were replenished with fresh media 2 h prior to lysis. Cells were then started on appropriate drug treatments 20 min–1 h prior to lysis (Torin1 1 h prior; Silvestrol 1 h prior; Sodium Arsenite 1 h prior; Puromycin 20–30 min prior). At the time of lysis, cells were washed in 1 mL PBS and 100 μL of Phos-Tag lysis buffer was added dropwise to the well (Phos-Tag lysis buffer recipe: RIPA (Thermo Fisher 89900), 3X Halt protease and phosphatase inhibitor cocktail (Thermo Fisher 78444), ≥40 U/mL benzonase (MilliporeSigma E1014), 1 mM TCEP). Cells were scraped directly from the plate in lysis buffer, gently pipetted to homogenize, transferred to ice for 10 min, and clarified by centrifuging at $8000 \times g$ at 4 °C for 5 min. Total protein concentrations of clarified lysates were determined by BCA assay (Thermo Fisher 23225), and lysates were resuspended to an equal concentration (100–500 ng/μL) in 1X Laemmli buffer and boiled at 95 °C for 5 min.

For LARP1 immunoblots, 3–5 μg of total protein was loaded per well, and samples were resolved by 5 or 6% SDS-PAGE containing 25 μM Phos-tag reagent (Wako, AAL-107) and 50 μM $MnCl_2$. Gel electrophoresis was performed in 1x Tris/glycine/SDS running buffer (100–125 V for 2–4 h), and an EDTA-free pre-stained protein marker (Apex Bio F4005) was used as a ladder. Gels were washed twice in 1x transfer buffer (25 mM Tris, 192 mM glycine, 10% v/v methanol) supplemented with 1 mM EDTA, followed by two washes in 1x transfer buffer without EDTA (10 min per wash). Gels were transferred to a PVDF membrane (Bio-Rad 1704273) in 1x transfer buffer (35 V, overnight, 4 °C). The next day, membranes were blocked in 5% non-fat milk (Santa Cruz Biotechnology sc-2325) w/v in TBST for 1 h and blotted using LARP1 primary antibody at a 1:1000 dilution in 5% milk in TBST at 4 °C overnight. The next day, membranes were washed $3 \times 10$ min in TBST at room temperature and hybridized with mouse anti-rabbit IgG-HRP secondary antibody (sc-2357) at 1:5000 dilution for 1 h. After 1 h, membranes were washed $3 \times 10$ min in TBST at room temperature and washed $1 \times 10$ min in TBS. Blots were developed using SuperSignal West Femto Maximum Sensitivity Substrate (Thermo Fisher 34095) using a ChemiDoc imaging system (Bio-Rad).

## Data availability

All study data are included in the article and/or supporting information. Any new materials or reagents generated in this study are available upon request. Nanopore sequencing data is deposited in the Gene Expression Omnibus (GSE246077; https://www.ncbi.nlm.nih.gov/geo/query/acc.cgi?acc=GSE246077). All cryo-EM maps and molecular models generated in this study have been deposited in the Electron Microscopy Data Bank (EMDB) and in the Protein Data Bank (PDB) with accession codes: EMD-38548 (https://www.ebi.ac.uk/emdb/EMD-38548) and PDB 8XP2 (https://doi.org/10.2210/pdb8XP2/pdb) (LARP1-40S, PYM1 sample); and EMD-38550 (https://www.ebi.ac.uk/emdb/EMD-38550) (LARP1-40S, LARP1 sample). Code for analyzing sequencing and qPCR data and for generating plots has been posted on GitHub at repository 2023_Saba_Larp1 (https://github.com/jakesaba/2023_Saba_LARP1).

The source data of this paper are collected in the following database record: biostudies:S-SCDT-10_1038-S44318-024-00294-z.

## Peer review information

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

## Acknowledgements

We thank members of the Green, Cheng, and Timp Labs for their helpful comments and suggestions during the preparation of this manuscript. We thank Paul Hook for help with Nanopore sequencing analysis. We thank Carson Thoreen for providing LARP1-KO cell lines (HEK293T) and LARP1 expression plasmids. We thank the Center of Cryo-Electron Microscopy at Fudan University for technical support. This research was supported by grants from the National Key R&D Program of China (2023YFC2413204), National Natural Science Foundation of China (32371350) and Shanghai Municipal Science and Technology Commission grants (22410712400, 22ZR1413600) to JC; National Institutes of Health (NCI, F30CA260910) to JAS; National Institutes of Health MSTP program (T32GM136577) to JAS and KLS; National Institutes of Health (NHGRI, HG010538) to WT. RG acknowledges support for this research from the Howard Hughes Medical Institute.

## Author contributions

**James A Saba**: Conceptualization; Data curation; Formal analysis; Validation; Investigation; Visualization; Methodology; Writing—original draft; Writing—review and editing. **Zixuan Huang**: Data curation; Formal analysis; Investigation; Visualization; Methodology; Writing—review and editing. **Kate L Schole**: Investigation; Methodology; Writing—review and editing. **Xianwen Ye**: Investigation; Methodology; Writing—review and editing. **Shrey D Bhatt**: Investigation; Writing—review and editing. **Yi Li**: Resources; Methodology; Writing—review and editing. **Winston Timp**: Resources; Software; Funding acquisition; Methodology; Writing—review and editing. **Jingdong Cheng**: Conceptualization; Data curation; Supervision; Funding acquisition; Investigation; Visualization; Methodology; Writing—review and editing. **Rachel Green**: Conceptualization; Supervision; Funding acquisition; Investigation; Methodology; Writing—original draft; Writing—review and editing.

Source data underlying figure panels in this paper may have individual authorship assigned. Where available, figure panel/source data authorship is listed in the following database record: biostudies:S-SCDT-10_1038-S44318-024-00294-z.

## Disclosure and competing interests statement

RG is a member of the Advisory Editorial Board of The EMBO Journal. This has no bearing on the editorial consideration of this article for publication. RG is on the scientific advisory board of Alltrna, Initial Therapeutics, and Arrakis Pharmaceuticals and serves as a consultant for Vertex Pharmaceuticals, Bristol-Myers Squibb (Celgene), Monta Rosa Therapeutics, and Flagship Pioneering. RG previously served on the scientific advisory board at Moderna. WT has two patents (8,748,091 and 8,394,584) licensed to ONT and received reimbursement for travel, accommodation, and/or conference fees to speak at events organized by ONT.

