## [Peer Review File · The EMBO Journal]

LARP1 binds ribosomes and TOP mRNAs in repressed complexes

James Saba, Zixuan Huang, Kate Schole, Xianwen Ye, Shrey Bhatt, Yi Li, Winston Timp, Jingdong Cheng, and Rachel Green

Corresponding author(s): Rachel Green (ragreen@jhmi.edu) , Jingdong Cheng (cheng@fudan.edu.cn)

Review Timeline:

Submission Date:	16th Aug 24
Editorial Decision:	2nd Oct 24
Revision Received:	13th Oct 24
Accepted:	16th Oct 24

Editor: Cornelius Schneider

Transaction Report:

The first review round of this manuscript was performed in another journal.

Point-by-point Reviewer responses:

Reviewer: 1

As summarized, it has been long known that the 5' Terminal OligoPyrimidine motif of the TOP-mRNAs, is the regulatory element through which their translation is tightly coordinated with demand for protein synthesis at the cellular level, as these ~100 mRNAs encode the ribosomal proteins and a number of abundant translation factors required for protein synthesis including PABP, several elongation factors and RACK1. Tight control includes swift negative translational regulation in response to nutrient shift-mediated or drug-induced mTOR inhibition into a state that preserves readiness to resume translation upon return of favorable conditions. Nearly a decade ago, LARP1 was identified as the major candidate trans-acting factor responsible. It serves as a robust mTOR-responsive, negative regulator of translation of mRNAs with the 5'TOP, which is recognized by its C- terminal DM15 motif. TOPs can comprise 10-20% of cellular mRNA. From a separate line of investigations, LARP1 is recognized as an important factor over-expressed in certain cancers in which it promotes expression of an associated transcriptome of non-TOP mRNAs, via signaling that coordinates protein synthesis and proliferation by pathways dependent on as well as independent of mTOR. Yet, the activities and structure-function aspects of this multidomain protein and their places in the control of TOP-mRNAs by mTOR which coordinates major signaling pathways, as well as in other signal pathways that control cellular proliferation, are far from fully understood.

The authors provide compelling evidence that LARP1 binds to nontranslating 80S ribosomes, well supported by cryo-EM structures (Fig. 2) in which a region of LARP1 with no prior known function, designated RBR extends as an α -helix into the mRNA binding channel of the ribosome (like the SARS-CoV-2 NSP1 protein element). The structure is supported by eCLIP data (ref 36) that reports crosslinks of LARP1 to the precise region of 18S rRNA in contact with RBR in the cryo-EM structure. Functional relevance of the structure is supported by experiments using a mutagenized LARP1 protein containing nine alanine substitutions in two segments of the RBR. When expressed in LARP1-KO cells the mutated protein completely disrupted LARP1 ability to shift TOP-mRNA to 80S complexes. LARP1-TOP-mRNA shifts from polysomes to 80S complexes associated with conditions of mTOR inhibition has been noted by and characterized by others and is an important part of this study (Fig 1). The authors provide evidence that the TOP-mRNA-LARP1-RBR-80S structure represents vacant 80S complexes in which mRNA is not occupying the decoding center of the ribosome. In-depth analysis revealed that 48 hours of mTOR inhibition led to translationally inactive LARP1-associated TOP-mRNAs on ribosome complexes, and that these were restored to translating polysomes in a "loaded gun" fashion upon reversal of inhibition (ref 21). Interesting was that 48 hr. mTOR inhibition led to LARP1- associated TOP-mRNAs on 40S complexes. Fig. S4 shows three segments of the RBR of the LARP1-40S complex, providing a molecular basis for the findings in refs 21 and importantly ref 34.

A weakness is apparent absence of link to the mTOR-pathway of LARP1-function which is complicated and emphasized by results of the crude Phos-Tag assay.

We appreciate the generally supportive comments by the Reviewer regarding the quality of our work. While we understand the Reviewer's concern about the mTOR connection, we emphasize that we only claim that TOP-40S and TOP-80S formation do not absolutely depend on changes in mTOR signaling. We believe our data provide strong evidence for this claim. We do not dispute the many studies that show that mTOR signaling is a strong driver of TOP regulation (PMIDs 25940091, 29244122, 33398329, 34818049, to name a few).

The 1096 aa LARP1 protein is phosphorylated at 26 mTOR-sensitive sites (ref 33) which lead to a large mobility shift in response to Torin although LARP1 is also known to be phosphorylated independent of mTOR. The authors should consider if by comparison to Torin, a small but appreciable mobility shift relative to the control lanes leave open a possibility that some conditions may alter phosphorylation of many fewer sites (refs 24-25) and be associated with their observed effects?

Absolutely. We acknowledge that smaller (harder to visualize) changes in LARP1's phosphorylation status could mediate the observed effects. We have revised the text to be more inclusive of this possibility with the following statement: "While we cannot exclude the possibility that small-scale phosphorylation changes to LARP1 mediate TOP-80S formation, these data show that global phosphorylation of LARP1 is not appreciably decreased in these regimes."

Also, as noted in point 1 below, they should determine if each condition is LARP1-dependent.

Addressed in response to "Specific Issue" 1 below.

These data suggest the paper will appear to not meet the functional relevance significance suggested by the Title, which to many readers will imply response to mTOR signaling.

We did not intend our original title to imply a response to mTOR signaling. We have revised the title and believe that this new title also does not imply a response to mTOR signaling.

New title: "LARP1 binds free 40S subunits to regulate its own translation."

Specific issues:

- 1. The experiments that test if depleting diverse components of the translation initiation machinery, eIF4A, eIF3B, eIF2S1, and other means to increase free ribosomes, as analyzed in figures 4B-F: These analyses represent much work but did not test if their result to shift TOPs to 80S complexes were LARP1-dependent, a critical feature of the model that should be demonstrated for each condition tested. Although results in fig. S6B showed*

distribution of LARP1 protein shifted to 80S fraction, this was only for one instance, upon eIF4E knockdown. And although LARP1-KO cells fail to shift TOPs to 80S, that was in response to Torin (Fig 2K, S1B) a different condition that was to be distinguished for the purposes tested here. Western blots showing LARP1-shift for each of the conditions should be shown, and whether the shift from polysomes to 80S occurs or not in LARP1 KO cells. Without such evidence the experiments do not prove the hypothesis that the conditions that increase free ribosomes by the means tested all use the LARP1-dependent mechanism proposed. There could be another mechanism.

We appreciate the Reviewer's concern here. To address this concern, we more broadly tested for formation of the TOP-80S in LARP1-KO cells. For all of the conditions tested, LARP1-KO cells fail to form the TOP-80S (figs. S4A-C). We also used our spread, high-K⁺ gradient trick to show that all of these TOP-80S complexes involve the weakly-associated 40S and 60S subunits directly bound to (and dependent on) LARP1 (figs. S4D-E). These data clearly demonstrate that all of our conditions which increase free ribosomes rely on the LARP1-dependent mechanism for TOP-80S formation.

- 2. The authors provide evidence that support the view that this binding to nontranslating free 80S ribosomes would occur in a TOP-mRNA-dependent manner. Although this would be important in the regulatory model in which LARP1 inhibits translation when bound to TOP-mRNA depicted in Fig 4G, it is not substantiated in this paper. It has not been experimentally determined if LARP1 can bind non-translating free-80S ribosomes in which the translational inhibitory element the RBR occupies the mRNA channel in the absence of TOP-mRNA binding by LARP1.*

We agree. To address this point we constructed a triple mutant LARP1 construct wherein we introduced documented point mutations that disrupt the La, PAM2, and DM15 domains (termed "LPDmut"; PMIDs 35979957, 33292040, and 28379136). We transfected this construct into LARP1-KO cells and tested its ability to bind 80S ribosomes under Torin1 treatment. We find that this mutant robustly binds ribosomes even though it cannot form the TOP-80S due to loss of TOP binding (Fig. 4C). These data demonstrate that the RBR can bind ribosomes in the absence of known TOP mRNA-binding interactions by LARP1. Thus, TOP binding is not required for the RBR to bind in the mRNA channel of the 40S or 80S ribosome.

- The authors attempt to address this in the Discussion, "This view is supported by the observation that mutations in either the RBR or DM15 domains preclude TOP-80S formation (our data and (35)),” though no experiments in which DM15 mutants were used were found in this paper.*

Addressed in the point above.

- They state, “While our data do not allow us to assign an order of events for complex formation, we are inclined toward a model in which free 40S subunits first bind LARP1 and this induces binding to TOPs.” This should be better assimilated with data from Figure 1F from which the authors noted that the TOPs directed ordered subunit joining (page 5), which seems a different message.

As noted above, our data now clearly demonstrate that ribosome binding does not require TOP binding. However, in terms of ordered subunit joining to form the TOP-80S complex, we have shown and now clearly stated within the manuscript that 40S binding occurs before 60S binding.

- The authors could strengthen this by doing qPCR of the same RNA samples in Figure 1F for non-TOP mRNAs.

As requested, we have performed qPCR for the non-TOPs Ubi5 and Ndufa1 from the same samples presented in Fig. 1F. Under Torin1 conditions, non-TOP distributions are largely unaffected by these knockdowns (fig. S2F).

3. Issues regarding information about the structure. The RBR is depicted in figure 2A as region 650-730 yet the text describes “the first segment of residues 660-694” consistent with its resolution in figure 2D,E beginning with residue 660. If the RBR is to be designated 650-730, where are the first 10 residues?

Agreed. We have updated the RBR domain in all figures and text to only include residues 660-724 for which we have defined LARP1 density.

- Information about density associated with RACK1 leaves uncertainty as to what conclusions can be drawn. Evidence that the density should be attributed to LARP1 vs. some other associated protein of unknown identity should be presented. Instead, the text descriptions appear to assume it is LARP1 and the exact region of the RBR is uncertain. This is important in its own right but would also be notable because LARP4 (and LARP4B) interact with RACK1 and share other functional features with LARP1.

We agree with the Reviewer that we should not assume that this density is LARP1. We have updated the text to simply state that there is additional density near RACK1 in our structures which we are unable to assign. This density could be a third domain of the RBR or a different protein altogether.

- What is the interaction surface area of the RBR with the 80S monosome? (see point A below).

While we do not have a structure of the RBR with the 80S monosome, we can calculate the interaction surface area with the 40S subunit. According to the PDBePISA server, the interface

area between the RBR and the 40S subunit is approximately 2673 square angstroms (\AA^2). We have updated the text to discuss this estimated interaction surface area.

4. *The relevance of data on LRRC47 is presently unclear.*

Can the authors determine what fraction of cellular ribosomes contain both LARP1 and LRRC47?

Does LRRC47 bind cytoplasmic 80S ribosomes in LARP1-KO cells? Is there evidence of functional relevance of finding both proteins together. Where do LARP1, LRRC47 and other ribosomal proteins colocalize?

This data doesn't seem to fit well.

We agree with these points by the Reviewer(s). To address these points and those of the other Reviewers (who also thought LRRC47 was disruptive) we have removed the section on LRRC47 and all associated data and discussion. This is an overall improvement to the manuscript.

Other points

A. *Fig 1E is difficult to interpret because the rRNA EtBr gel is not shown. According to the lower A260 tracing in High-K⁺, one would expect that the peak of 18S and 28S rRNAs would be in fraction 14 (according to numbering under LARP1 western below) while the peaks of Rpl39 and Rpl21 are in fractions 12 and 11 respectively. Showing the EtBr for the rRNAs would be helpful.*

As requested, we have run EtBr gels using RNA from the same samples and included these below the plots in Fig. 1E. These data demonstrate that the 18S rRNA distribution mimics A260 absorbance on the gradient trace, as expected.

- *The point is that relative to the A260 tracing showing 80S (fraction 14) which presumably contains non-vacant monosomes, the tracing shows low absorbance in fractions 11-12, less than expected for en-masse shift of TOP-mRNAs which reportedly represent >10% of cellular mRNAs (there is no description of the inset for the A260 tracing in the legend).*

The Reviewer makes the point that the shift of TOPs (and associated LARP1-bound ribosomes) to fractions 11-12 should be apparent on the A260 tracing. We do not think this is true based on the following back-of-the-envelope calculation. TOPs account for 10-20% of cellular mRNAs and mammalian cells contain on average ~300,000 mRNAs (PMID 10581018), meaning there are ~50,000 TOPs per cell. In contrast, mammalian cells contain, on average, ~5 million ribosomes per cell (PMID 6853516 and PMID 889788). Therefore, if all TOPs shift with LARP1 and an associated ribosome, this would only represent a shift of ~1% of the cellular ribosome pool

(50,000 out of 5 million). As the Reviewer predicts, there is a small amount of 18S rRNA in fractions 11-12 in the EtBr gel, and it is reassuringly less than that observed in fractions 13-14 where the non-dissociated 80S peak runs.

- *An underlying issue is whether vacant 80S monosomes are stabilized against dissociation into 40S and 60S subunits when occupied by LARP1-TOP-mRNA, or are they destabilized relative to vacant 80S monosomes that are not bound by LARP1-TOP-mRNA?*

We are simply stating what we see in our experiment. We observe a left-shift of LARP1 and the TOP-80S (i.e. dissociation of complex during sedimentation) relative to the majority of 80S monosomes by A260 (Fig. 1E). However, the “canonical” A260 80S peak does not split until gradients are run with higher KOAc concentrations in the range of 300-500mM KOAc (see A260 traces in fig. S1D). These data suggest that LARP1-bound 80S monosomes are destabilized relative to the majority of 80S monosomes.

- *Do vacant 80S monosomes other than those occupied by LARP1-TOP-mRNAs exist? - where would they be? Their apparent absence suggests that detection of proposed vacant LARP1-TOP-mRNA-80S complexes indicates they are more stable than others. If LARP1-RBR stabilizes these complexes the data may provide theoretical support for a model as a negative element of translation that helps maintain readiness to resume translation.*

An underlying complexity raised by the Reviewer here (and above) is that other 80S species exist during mTOR inhibition beyond those bound by LARP1. For instance SERBP1 also binds in the mRNA channel of non-translating mammalian ribosomes during mTOR inhibition (PMIDs 36691768 and 32687489). We therefore blotted for SERBP1 across the same norm-K⁺ and high-K⁺ gradients to test whether SERBP1-bound monosomes are more resistant to salt treatment compared to LARP1-bound monosomes. In contrast to LARP1 which shifts left with the TOP-80S in High-K⁺ gradients, SERBP1 continues to sediment with the major 80S monosome peak (Fig. 1E). These data suggest that at least SERBP1-bound monosomes are more stable to salt-induced dissociation than LARP1-bound monosomes. In relation to the Reviewer’s last point, although we like the model that ribosome-binding by LARP1 on TOPs might advantage their return to the translating pool, we do not have data to support such a model. Instead, we cite this possibility with the following statement: “...rather these complexes could play a role in the re-initiation of TOP translation following mTOR activation.” (PMID 34818049)

- B. *A robust system for interrogating sedimentation of TOPs with 40S and 80S ribosomes in Fig. 1, as "TOPs redistribute en masse to the 80S fraction (Fig. 1B)" is a bit strong because it applies better to TOP redistribution determined by nanopore sequencing performed on U266B1 cells (Fig. 1B), than to most subsequent experiments with HEK293T cells. While*

RACK1 mRNA, in Fig 1C (HEK293T) is a TOP with robust 80S formation, other TOPs in Fig. 1D, 4C, 4E are less impressive than expected based on U266B1 results.

We appreciate the reviewer's question here and think this is a misunderstanding resulting from the fact that we gathered fewer fractions for non-spread than spread gradients. If one compares the y-axes of the plots in Figs. 1B/1C (non-spread gradients) to Figs. 1D/S4D/S4E (spread gradients; originally Figs. 1D/4C/4E), then it might seem that the TOPs in 1D/S4D/S4E have less robust TOP-80S formation. However, the TOP-80S is distributed amongst more fractions in the spread gradients and one must sum fractions comprising the TOP-80S to get the total population. In Figs. 1D/S4D/S4E the cumulative Rpl39 population in the TOP-80S is between 35-70% of the total Rpl39 population. This is comparable to the 40-60% distribution of TOPs in the TOP-80S in Figs. 1B and 1C.

C. Figure 1F nicely shows distribution of two TOP mRNAs after Torin treatment of cells in which a 40S or a 60S subunit was depleted. The authors noted importantly that the TOPs directed ordered subunit joining. They should probe/qPCR the same RNA samples for non-TOP mRNAs and include in the same figure.

As discussed above, we have now probed for two non-TOPs (Ubl5 and Ndufa1) from the same samples presented in Fig. 1F. Under Torin1 conditions, non-TOP distributions are mostly unaffected by these knockdowns. We have included these figures as a supplemental panel in fig. S2F.

Reviewer: 2

In the manuscript, “LARP1 senses free ribosomes to coordinate supply and demand of ribosomal proteins,” Saba et al utilize cryo-electron microscopy to identify part of La-related protein 1 (LARP1) bound in the decoding center of the human small ribosomal subunit. Considering established support for LARP1 as a translational repressor of TOP mRNAs, these structures enable the hypothesis that LARP1 binds to the 40S to inhibit translation of TOPs. The team further tests their hypothesis with polysome profiling employing classical high-salt “spread” gradients, followed by RT-qPCR, under various conditions, including mTORC1 inhibition, knockdown of ribosomal proteins, knockdown of initiation factors, and induction of different stress responses. The data: 1) demonstrate that the LARP1-TOP complex interacts with a loosely-associated 80S, largely through the 40S; 2) define the ribosome binding region (RBR) domain of LARP1; 3) suggest that the association of LARP1-TOP complex with 40S occurs independent of mTORC1 signaling; and 4) TOP mRNAs redistribute within the gradient upon various treatments. This is an interesting and exciting manuscript that holds potential to reveal novel information about TOP mRNA translation regulation. The novelty of this study is demonstrating that part of LARP1 binds the decoding center of the ribosome. Additionally, the data is very clean. However, there are several issues that must be addressed:

Major

- *There are no statistics provided for most experiments. Experiments are shown with standard deviation between technical replicates. Why aren't the biological replicates plotted?*

We appreciate the Reviewer's concern about statistical rigor and reproducibility. To address this concern, we have provided biological replicate experiments for Figs. 1D, 1E, 1F, 2A, 2B, 2C, 2D, 4B, 4C, 6C, 6E, and 6F in the Supplementary materials (figs. S2A, S2C, S2E, S3B, S3D, S3F, S4F, S9A, S9B, S11E, S12B, and S12C, respectively) and noted these replicates in the figure legends. We have also provided data from multiple biological replicates or multiple cell lines in the same figure when possible (Figs. 5A, 5B, 6A, 6B, 6C, S10A, S10B, S11A, S11B, S11C, S11D, and S11E). As we hope the Reviewer will appreciate, the data (and conclusions therein) are highly reproducible. We have avoided combining biological replicates for sucrose gradient qPCRs because we find that day-to-day variability in the sedimentation profiles or in the non-equivalent number of fractions taken erodes the richness of the data.

- *The data linking LARP1 to sensing free ribosomes is correlative, at best. Therefore, the title should be revised. Further, “sensing” implies that information is transferred somewhere, but no data is provided to suggest what LARP1 does with this information.*

We agree that use of the word “sensing” was not substantiated in our first submission of the manuscript. By contrast we feel that our data now substantiates the model that LARP1 senses free 40S subunits to regulate its own protein levels accordingly. We have revised the title in accordance with this model to “LARP1 binds free 40S subunits to regulate its own translation”.

- *None of the gradients has accompanying LARP1 western blots, except for Figs. 1E, 2L, and S6; this information is essential for supporting the idea that various treatments result in the occupancy of the “loose” 80S by LARP1-TOP complex. It is especially important in Fig. S1C when higher salt causes TOPs to sediment even further left.*

As requested, we “western blotted” for LARP1 protein across gradients in fig. S2D (previously S1C). LARP1 sedimentation mirrors TOP distribution in these gradients, providing support for its association in these complexes.

To further address the concern that the TOP-80S complexes formed under the various treatment conditions are all “loose” 80S complexes bound directly by LARP1, we tested for formation of the TOP-80S in LARP1-KO cells. For all conditions, LARP1-KO cells fail to form the TOP-80S (figs. S4A-C). We also used our spread, high-K⁺ gradient trick to show that all of these TOP-80S complexes involve weakly-associated 40S and 60S subunits directly bound to (and dependent on) LARP1 (figs. S4D-E). These data demonstrate that the TOP-80S complexes formed under these conditions all correspond to the non-translating, loosely-associated 40S and 60S subunits described in Figure 1 of the manuscript.

- *No evidence for TOP protein expression/translation repression is provided in any condition. This is required to support statements like, “These data indicate that the TOP-80S under all of these conditions corresponds to non-translating, loosely-associated 40S and 60S subunits which are bound to LARP1 and not to the TOP itself.”*

This is an important point by the Reviewer and helped us to directly test and clarify our model from the previous submission. To directly measure TOP translation repression, we constructed rapidly degraded nano-luciferase reporters (nLuc-PEST) driven by the promoter and 5’UTR of either Rpl39 (TOP) or Ubl5 (non-TOP) (Figs. 5A and S10A). These reporters helped us to clarify that ribosome binding by LARP1 is not critical for TOP repression (though the ribosomal subunits are present in the repressive complex).

- *How do these treatments affect the health of the cells? Are they actively growing? This information is critical to understanding TOP metabolism.*

The translation factors that we knocked down are essential (PMID 26472758). Under our “knock-down” conditions, cells grow slowly but remain alive and attached to the plate. In our acute drug treatments (1h or less) cells are also alive and remain attached to the plate and we do not observe changes in cell morphology. Importantly, at longer time points, some of these treatments are

known to arrest or kill cells (PMIDs 19150980, 25079319, 25621764, 32435426); we were careful to avoid these conditions where cells are more unhappy.

- *The information in Fig. 1B needs more analysis: is there a length-dependence? Subcategories within TOPs? How does this information compare to that in Philippe et al and van den Elzen et al?*

As requested, we have performed a deeper analysis of the data in Figure 1B (figs. S1A-E). As expected, we did find that longer mRNAs tend to sediment deeper (either because of their intrinsic length or because their CDS is typically loaded with more ribosomes). We used this analysis to rationalize our use of very short mRNAs with an annotated length of < 500 bp for our spread gradients. The data shown in Fig. 1B compare nicely to Philippe et al. (PMID 32094190) who showed that TOPs are repressed upon mTOR inhibition and van den Elzen et al. (PMID 36441824) who showed that +1-C mRNAs are particularly sensitive to mTOR inhibition. Importantly, it is difficult to make direct comparisons to these datasets as their conclusions are drawn from ribosome footprint profiling data while ours are drawn from mRNA sedimentation across gradient data.

- *Figure 3 is nice, but is an abrupt interruption to an otherwise streamlined story. Additionally, the data in the top panel of Fig. 3C indicates that more replicates need to be performed.*

All Reviewers found the section on LRRC47 to be disruptive. As such, we have removed the section on LRRC47 and all associated data and discussion.

Minor

The labeling of the “loose” 80S complex needs attention. For example, in Fig. 1E, it is labeled as 60S, but the highlight indicates it is the 80S

We have tried to make this clearer in labeling and text. The “60” label is intended to label the 60S peak on the A260 plot while the highlight is intended to label the sedimentation of the TOP mRNA in the TOP-80S complex (as defined by qRT-PCR results). We have updated the figure legends to more clearly state these distinctions.

Figure 4b and S7 si-eIF2S1 typo on polysome figure

The typo has been corrected.

Polysome profiling qPCR for some published LARP1 targets like RPS6 or RPL32 should be included.

We appreciate this suggestion but feel that we have used multiple such targets in our analysis already. Three of the main TOPs that we follow are published LARP1 targets (RACK1, RPL5 and RPL11). Philippe et al. (PMID 29244122) specifically demonstrate that RACK1 mRNA (also called GNB2L1) is translationally repressed by Torin1 in a LARP1-dependent manner in Figure 1C of their paper. Additionally, Fuentes et al. (PMID 34818049) show that RPL5 and RPL11 mRNAs are protected from degradation upon mTOR inhibition in a LARP1-dependent manner in Figure 1A of their paper.

We further note that we intentionally probed for short TOPs (Rpl39 and Rps21) for our spread gradient analyses. While these particular mRNAs were not used in other publications, they have a consensus TOP motif at their 5' end and follow the expected distribution of TOPs in our sequencing data.

Reviewer: 3

This exciting manuscript from Saba et al. characterizes an unexpected complex between the 40S and 60S ribosomal subunits and the RNA-binding protein Larp1. Larp1 is best known as a post-transcriptional regulator of mRNAs with TOP motifs (TOP mRNAs) and is primarily controlled by the mTOR pathway. Larp1 was previously shown to interact with the 40S subunit, but this manuscript adds considerably more structural detail, including the discovery of a Larp1 domain that inserts into the 40S mRNA-binding channel. This interaction occurs independent of mTOR activity and drives the formation of translationally silent TOP-80S complexes. These claims are well-supported by the data presented. Based on these observations, the authors propose a model whereby Larp1 senses the concentration of free 40S subunits and responds through post-transcriptional regulation of TOP mRNAs, which encode nearly all ribosomal proteins. The primary unanswered question here is whether the Larp1-40S interaction directly impacts Larp1 functions in the translation or stability of TOP mRNAs, or whether this interaction serves some separate function (e.g. stabilization of free 40S subunits, 40S biogenesis, etc.). Comments are detailed below:

We thank the Reviewer for their supportive comments regarding the quality of our work. Importantly, the Reviewer recognized a major gap in knowledge from our previous submission which is that we had not nailed down the function of ribosome binding by LARP1. Addressing this Reviewer's comments was instrumental in developing the more complete story that we now have.

- 1. This manuscript implies a mechanistic connection between the formation of the TOP-80S complex and Larp1 translation and stability function but doesn't test it directly. This could be done relatively easily using the RBR mutant allele described in Figure 2K. Does this mutant still bind TOP mRNAs, stabilize them, or repress their translation? Evidence of functional connection between the TOP-80S and TOP regulation would strengthen the hypothesis that Larp1 links detection of free 40S subunits with the production of ribosomal proteins.*

To address these concerns, we developed reporter assays to look at translational repression and used qPCR to evaluate steady state TOP mRNA levels. Surprisingly, we now show that the RBRmut allele of LARP1 translationally represses and stabilizes TOPs to an equal or greater extent than the WT allele (Figs. 5A-B and S10B). Therefore, we no longer believe that LARP1-ribosome binding is critical to TOP repression or stabilization *per se*, though ribosomes are clearly found in the repressive LARP1-TOP complex. Instead we have discovered that ribosome binding allows LARP1 to coordinate its own levels (and thereby the translational regulation and abundance of TOPs) to the abundance of free 40S subunits. Moreover, we speculate that ribosome binding to the TOP-40S or TOP-80S could be critical for other features of this complicated control system, such as facilitating re-initiation of translation when stress is relieved as proposed in PMID 34818049.

2. *Results in Figure 4 argue that essentially any perturbation to the translation process that increases free 40S and 60S subunits will drive the formation of TOP-80S complexes. This suggests TOP translation and/or stability is also selectively affected by these conditions, but this is not necessarily the case. For example, arsenite triggers TOP-80S formation (Figure 4D,E), but previous translation profiling studies of the integrated stress response don't show a selective decrease in TOP translation (Sidrauski et al., 2015. PMID: 25719440). Similar to the previous comment, the authors should test whether repressing translation downstream of eIF4F still selectively affects the translation or stability of TOP mRNAs.*

The Reviewer was correct in asserting that our previous data did not mandate that TOPs are preferentially sensitive to perturbations to the translation process. To directly test this, we generated TOP and non-TOP reporters (Fig 5A and fig. S10A). Importantly, while the TOP reporter was more sensitive than the non-TOP reporter to translation inhibition by Torin1, it was less sensitive to Silvestrol, and equally sensitive to puromycin at a range of concentrations (fig. S10A). While we also tested sodium arsenite in this assay, the treatment disrupted the FFLuc measurement (normalization control) and made those data untrustworthy. Nonetheless, these data together show that simply increasing free ribosomes does not drive greater TOP repression relative to non-TOPs. Importantly, however, the TOPs are repressed in each case and we see the correlated appearance of the LARP1-bound TOP-80S repressive complex.

3. *The findings described here raise many questions about how the Larp1-40S interaction is formed and also released. While many of these could be explored in the future, it would be interesting to know at least whether the RBR is sufficient for this interaction, or whether other domains (La/RRM, DM15) are also involved. This could be easily tested by using sucrose gradients to test the migration of mutant alleles, as in Figure 2L.*

See response to Reviewer 1, specific issue 2. Briefly, we introduced point mutations that disrupt the La, PAM2, and DM15 domains of LARP1. When we introduce this mutant construct to LARP1-KO cells, we find that it continues to bind ribosomes even though it cannot form the TOP-80S due to loss of TOP mRNA binding (Fig. 4C). These data demonstrate that the RBR can bind ribosomes in the absence of known mRNA-binding interactions by LARP1.

4. *PYM1 seems like an unusual choice for purifying ribosomes, as it also has reported functions in recruiting specific mRNAs to the translation machinery. Could the authors comment on this choice, and whether anything known about its function might have enriched for Larp1-40S complexes?*

While PYM1 is proposed to be involved in regulating the EJC during the pioneer round of translation, it has also been shown to directly interact with 40S ribosomes (PMIDs 18026120 and 19410547). Our initial goal in performing immunoprecipitations (IPs) on PYM1 was to capture

ribosomes participating in the pioneer round of translation. Unexpectedly, we observed the presence of LARP1 and LRRC47 in our PYM1 sample (LARP1 was the most abundant protein in the IP!). Importantly, we independently showed that a LARP1 IP enriches for LARP1-40S complexes that are indistinguishable from those in the PYM1 IP. While we do not know why the PYM1 IP enriched for LARP1-40S complexes, this unexpected finding could suggest a functional association between PYM1 and LARP1, which requires further investigation.

5. *It is unclear to me how important LRRC47 is for Larp1 function or TOP-80S formation. The authors suggest that LRRC47 might stabilize Larp1-40S complexes by interfering with 60S joining. Another hypothesis is that any perturbation that increases free 40S subunits will drive the formation of Larp1-40S complexes, similar to the Rpl31 knockdown in Figure 1F. The inclusion of LRRC47 results would seem more relevant if there was more evidence that LRRC47 impacted some function of Larp1. For instance, does knockdown of LRRC47 impact any other function of Larp1 (e.g. TOP binding)?*

All Reviewers found the section on LRRC47 to be disruptive. As such, we have removed the section on LRRC47 and all associated data and discussion.

6. *A recent report argued that TOP mRNAs in the 80S fraction are actively translated to maintain a baseline level of synthesis (Schneider et al., 2022. PMID: 36223745). In contrast, the polysome analysis of WT and Larp1 KO cells (Figure 2K) suggests that the vast majority of TOP mRNAs in the 80S peaks are associated with translationally inactive TOP-80S complexes. Can the authors estimate what proportion of TOP mRNAs in the 80S fraction are bound by actively translating ribosomes? Also, there is a small amount of non-TOP mRNA present in the 80S fraction in Norm-K⁺ gradients (Figure 1D). Does this remain in the 80S fraction in High-K⁺ gradients, suggesting that it is actively translated, or does it shift with the TOP-80S?*

These are excellent questions.

To the first point, our salt titration suggests that few TOPs in the 80S fraction are bound by an actively translating monosome because the vast majority of the TOP-80S shifts left in High-K⁺ gradients. Based on these experiments we can roughly estimate that fewer than 10% of TOPs in the 80S peak are bound by an actively elongating monosome under Torin1 treatment. Nonetheless, we expect that there is a minor population of TOPs that have a single elongating ribosome on them and we speculate that this population was sequenced in Schneider et al. (PMID 36223745). Because ribosome footprint profiling only visualizes mRNA footprints protected by ribosomes, the much larger population of ribosomes bound directly to LARP1's RBR (where the ribosomes are not protecting the mRNA) were invisible to this analysis.

To the second point, as requested, we repeated the high-K⁺ titration for the non-TOP Ubl5 under Torin1-treatment. Somewhat surprisingly, we observed that a small population of 80S-

associated Ubl5 also shifts in high-K⁺ gradients (compare black and blue qPCR traces below). The relative fraction shifted for Ubl5 is much smaller compared to that for TOPs (i.e. in Fig. 1E essentially the total TOP population shifts). Additionally, the formation of this left-shifted Ubl5 peak was LARP1-dependent (compare blue and orange qPCR traces below). These data suggest that LARP1 has some low propensity to also bind non-TOPs during Torin1 treatment and that these LARP1-nonTOP complexes contain a loosely-associated 80S ribosome bound to LARP1's RBR. We note that previous in vitro studies show that the La/PAM2 domains bind polyA sequences and polyA-binding protein, respectively, as well as polyG and TOP 5'UTRs in a cap-independent manner (PMIDs 35979957, 33292040, 31601159); these diverse specificities may explain why we observe a small amount of non-TOP binding in the LARP1-80S peak. For simplicity, we have opted not to include these data in the manuscript.

Dear Dr Green,

Thank you for submitting a revised version of your manuscript which was previously reviewed as another journal. We have consulted with an arbitrating referee who has reassessed the manuscript and thinks that all previous concerns have been addressed convincingly and which recommends publication of the manuscript in its current form. There remain only a few mainly editorial points that have to be addressed before I can extend formal acceptance of the manuscript:

- Please provide the manuscript file in .doc file without figures, and place the figure legends at the end of the ms file, below the References
- On the abstract page of the manuscript, please include 4-5 general keyword terms to enhance searchability.
- Please rename the conflict of interest section into "Disclosure and competing interests statement", and add the following disclaimer: "Rachel Green is a member of the Advisory Editorial Board of The EMBO Journal. This has no bearing on the editorial consideration of this article for publication." (for the authors tagged as EBM, EPAB, Council in eJP)
- As we are switching from a free-text author contribution statement towards a more formal statement based on Contributor Role Taxonomy (CRediT) terms, please remove the present Author Contribution section and instead specify each author's contribution(s) directly in the Author Information page of our submission system during upload of the final manuscript. See <https://casrai.org/credit/> for more information.
- There is a callout for "Data S1" which is not present in the current submission.
- Please provide either a "Yes" or a "Not Applicable" answer to each one of the questions in your Author Checklist (<https://www.embopress.org/pb-assets/embo-site/EMBO%20Press%20Author%20Checklist-1642513524327.xlsx>). In the last column of this checklist, only the sections of the manuscript where the relevant information can be found should be listed (the information per se should be included in the main manuscript file).
- All main and EV figures need to be uploaded as individual files with sufficient resolution/quality for production.
- Please rename the supplementary materials into a single APPENDIX PDF, headed by a brief Table Of Contents with page numbers of the included items. The nomenclature should be throughout the Appendix PDF and manuscript file: Appendix Figure S1-S10 and Appendix Table S1, with the appropriate callouts
- Please provide the Reagent and Tools Table. For more information, please check <https://www.embopress.org/page/journal/14602075/authorguide#structuredmethods> and download the template for Reagent Table (https://www.embopress.org/pb%2Dassets/embo-site/Reagents_Table_TEMPLATE.docx)
- Please provide suggestions for a short 'blurb' text prefacing and summing up the conceptual aspect of the study in two sentences (max. 250 characters), followed by 3-5 one-sentence 'bullet points' with brief factual statements of key results of the paper; they will form the basis of an editor-written 'Synopsis' accompanying the online version of the article. Please also provide an altered synopsis image, making sure that the aspect ratio conforms to our website's format - it should be exactly 550 pixels wide and between 300-600 pixels high.
- Please provide the specific URLs for datasets GSE246077, EMD-38549, EMD-38548 and PDB 8XP2, PDB 8XP3 and EMD-38550 in the data availability statement.
- Please define the box plots in terms of minima, maxima, centre, bounds of box and whiskers, and percentile in the legends of figures 6A, B.
- Please define the measure of center for the error bars in the legends of figures 1A, C; 2A-C; 3A-C; 5C.
- Please move the Supplementary Materials section to the appendix file.
- Please adjust the order of the manuscript sections: Title page with complete author information, Abstract, Keywords, Introduction, Results, Discussion, Methods, Data Availability Section, Acknowledgements, Disclosure and Competing Interests Statement, References, Main figure legends, Tables, Expanded Figure Legends.
- Please make sure to provide all the requested Source data files listed in the uploaded and attached Source Data checklist file, which you had been sent by my colleague Hannah Sonntag. Please complete the Source Data checklist and upload it to our online system. Source data files need to be saved in a scheme one figure/folder and then uploaded as .zip files. E.g. all the

Source data files for figure 1 need to be saved in a single folder and this needs to be zipped and then uploaded as "SD figure 1.zip" file.

With best regards,

Cornelius Schneider

Cornelius Schneider, PhD
Editor | The EMBO Journal
c.schneider@embojournal.org

We realize that it is difficult to revise to a specific deadline. In the interest of protecting the conceptual advance provided by the work, we recommend a revision within 3 months (31st Dec 2024). Please discuss the revision progress ahead of this time with the editor if you require more time to complete the revisions. Use the link below to submit your revision:

Referee #1:

Cell stress triggers the formation of a complex between 80S ribosomes and translation-related mRNAs that is observed in

diverse species from plants to mammals. The formation of this complex coincides with the translation repression and stabilization of these mRNAs, which has suggested a functional relationship. This manuscript from Saba et al. provides valuable insight into the mechanisms underlying this phenomenon. Studies over the last decade have shown that the RNA-binding protein Larp1 is required for the formation of these complexes, in part through recognition of terminal oligopyrimidine (TOP) motifs that are found on targeted mRNAs. However, the molecular basis of the TOP-80S complexes and whether their formation is required for stabilizing or repressing the translation of TOP mRNAs has been unclear. In this manuscript, the authors show that the TOP-80S complex is translationally inactive and forms in response to essentially any perturbation that increases the concentration of free ribosomal subunits. They further discover using cryo-EM that a region of Larp1 slots directly into the mRNA-binding channel of the small ribosomal subunit, and that this interaction is necessary for the formation of the TOP-80S complex. Surprisingly, and contrary to expectations in the field, this interaction is dispensable for the selective stabilization or translation repression of mRNAs with TOP motifs. Overall, this work provides a clear molecular understanding for how Larp1 drives the formation of 80S-TOP complexes while opening new questions about the function of the conserved Larp1-40S interaction. I recommend that this manuscript be accepted for publication.

Reviewer 1Review:

Cell stress triggers the formation of a complex between 80S ribosomes and translation-related mRNAs that is observed in diverse species from plants to mammals. The formation of this complex coincides with the translation repression and stabilization of these mRNAs, which has suggested a functional relationship. This manuscript from Saba et al. provides valuable insight into the mechanisms underlying this phenomenon. Studies over the last decade have shown that the RNA-binding protein Larp1 is required for the formation of these complexes, in part through recognition of terminal oligopyrimidine (TOP) motifs that are found on targeted mRNAs. However, the molecular basis of the TOP-80S complexes and whether their formation is required for stabilizing or repressing the translation of TOP mRNAs has been unclear. In this manuscript, the authors show that the TOP-80S complex is translationally inactive and forms in response to essentially any perturbation that increases the concentration of free ribosomal subunits. They further discover using cryo-EM that a region of Larp1 slots directly into the mRNA-binding channel of the small ribosomal subunit, and that this interaction is necessary for the formation of the TOP-80S complex. Surprisingly, and contrary to expectations in the field, this interaction is dispensable for the selective stabilization or translation repression of mRNAs with TOP motifs. Overall, this work provides a clear molecular understanding for how Larp1 drives the formation of 80S-TOP complexes while opening new questions about the function of the conserved Larp1-40S interaction. I recommend that this manuscript be accepted for publication.

Response:

We thank the Reviewer for their thorough, coherent, and concise review of our manuscript. This Review completely captures the contextual framework of the field and the advance which our study provides. We are excited that the Reviewer recommends publication of our work.

Dear Prof. Green,

I am pleased to inform you that your manuscript has been accepted for publication in the EMBO Journal.

Yours sincerely,

Cornelius Schneider, PhD
Editor
The EMBO Journal
c.schneider@embojournal.org
